# GaussTrace: Provenance Analysis of 3D Gaussian Splatting Models with Evidence-based LLM Reasoning

Haoliang Han [1]   Ziyuan Luo [1]   Renjie Wan [1]

## Abstract

3D Gaussian Splatting (3DGS) is a powerful technique for creating high-fidelity 3D assets. However, the widespread sharing and iterative modification of 3DGS models across digital platforms create pressing challenges for intellectual property protection and forensic traceability. To address this, we propose GaussTrace, a novel framework for constructing directed provenance graphs for 3DGS models. GaussTrace formulates provenance analysis as an evidence-based reasoning problem. It builds upon attribute-wise statistical profiling of 3DGS parameters to capture intrinsic properties. Moreover, we introduce hypothesis-driven editing simulations of common operations to provide auxiliary evidence for plausible transformation pathways. These statistical and simulated cues jointly enable a Large Language Model (LLM) to perform structured Chain-of-Thought (CoT) reasoning, yielding directional provenance inferences and explainable edge reasons. Experimental results demonstrate that GaussTrace effectively constructs evolutionary relationships among diverse 3DGS models, delivering accurate, interpretable, and robust provenance graphs without requiring model training or access to editing histories. Project page: https://haolianghan.github.io/GaussTrace.

## 1. Introduction

3D Gaussian Splatting (3DGS) (Kerbl et al., 2023) models are increasingly adopted in commercial and collaborative workflows (Bao et al., 2025). This widespread use makes it critical to ensure traceability of these models across multi-party distributions. To address this, we develop a novel

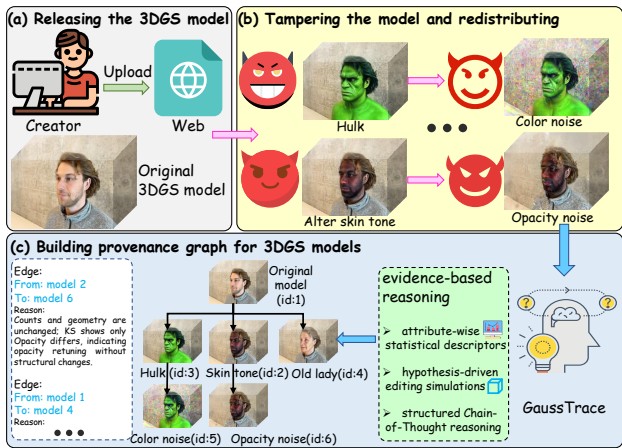

*Figure 1.* Illustration of our proposed scenario. A creator releases an original 3DGS model online, which may be modified and redistributed by malicious actors, leading to intellectual property infringement or harmful content propagation. Our GaussTrace can construct the evolutionary relationships among these models by performing evidence-based reasoning. The resulting provenance graph enables traceability of 3D assets for digital forensics.

framework that traces evolutionary lineages among 3DGS models with multiple modification histories.

In our scenario, illustrated in Figure 1, a creator releases an original 3DGS model online. Malicious users can then modify this model and redistribute the altered versions across the web. These modifications are often made and redistributed by multiple independent parties. As these variants distribute, they can infringe on the creator's intellectual property or be exploited to spread harmful content. Capturing the evolutionary relationship can help track how 3D assets are modified and redistributed, which supports accountability and prevents unauthorized edits.

In digital forensics, the task of constructing such evolutionary relationships is known as *provenance analysis* (Moreira et al., 2018). As shown in Figure 1, this process aims to build a directed graph, where nodes represent asset versions and edges denote modification relationships. Existing provenance analysis methods (Bharati et al., 2017; Moreira et al., 2018; Bharati et al., 2019; 2021; Zhang et al., 2025) have established a trace-based pipeline: 1) pairwise comparison

---

[1]Department of Computer Science, Hong Kong Baptist University, Hong Kong SAR, China. Correspondence to: Renjie Wan <renjiewan@hkbu.edu.hk>.

*Proceedings of the $43^{rd}$ International Conference on Machine Learning*, Seoul, South Korea. PMLR 306, 2026. Copyright 2026 by the author(s).

of digital contents to quantify the manipulation traces, and 2) graph construction based on the identified traces. However, existing provenance analysis methods are designed for images and cannot directly apply to 3DGS models due to fundamental structural differences. Though 2D images can also be rendered from 3DGS model, such rendering discards critical 3D structural information that reflect editing intent. Directly using existing provenance analysis methods (Moreira et al., 2018; Bharati et al., 2019; Zhang et al., 2025; 2020) on such rendered 2D images yields ambiguous outcomes, as shown in Figure 3. Thus, a new system should be developed to directly conduct provenance analysis on 3DGS models.

The challenges of conducting 3DGS provenance come from two factors. **First,** publicly released 3DGS models undergo modifications by multiple entities in varying sequences. This produces extended modification chains with complex correlations. **Second,** even a single modification typically affects multiple attributes within the 3DGS model simultaneously. These cross-attribute effects propagate through modification chains and compound trace obfuscation. The interaction between these two factors obscures essential provenance traces and impedes trace-based provenance analysis.

To address these challenges, we establish a new paradigm for provenance analysis. This paradigm departs from conventional trace-based pipelines and formulates provenance as an evidence-based reasoning problem. Provenance reasoning captures invariant dependencies between entities and identifies what determines an outcome. For 3DGS assets, this distinction is fundamental: when one model derives from another, the source imposes structural constraints on the derived model's properties regardless of which editing operations were used. These constraints manifest as detectable patterns in Gaussian attributes and spatial distributions, enabling reliable provenance inference without trace reconstruction.

Large Language Models (LLMs) provide a natural foundation for such provenance reasoning, given their demonstrated capabilities in structured inference (Wei et al., 2022). However, 3DGS models cannot be directly processed by LLMs, which operate on natural-language tokens. To make 3DGS representations comprehensible to the LLM, we introduce two key designs to extract evidence. First, we develop an attribute-level analysis that isolates the coupled effects of editing operations. This approach captures how each 3DGS attribute responds to modifications independently, disentangling composite signals into distinct, interpretable channels. Second, building on these disentangled signals, we construct evidential inputs by representing each 3DGS model through attribute-wise statistical descriptors combined with simulated editing outcomes. These components constitute a unified evidence basis for reasoning based on LLM.

We present our whole framework **GaussTrace** in Figure 2, a novel framework for constructing directed provenance graphs of 3DGS models. We first compute attribute-wise statistical profiles to capture intrinsic properties across 3DGS parameters. Furthermore, we introduce hypothesis-driven editing simulations, providing a semantic vocabulary of plausible operations against which observed shifts can be interpreted. Moreover, GaussTrace employs a large language model to perform structured Chain-of-Thought reasoning over this combined evidence, enabling it to infer directional relationships and generate natural-language justifications for each provenance edge. Our key contributions can be concluded as follows:

- The first framework for constructing directed provenance graphs of 3DGS models, which ensures model security throughout multi-party distribution.

- A novel approach that extracts LLM-interpretable evidence directly from 3DGS models and performs structured reasoning to infer provenance relationships.

- Extensive evaluations demonstrating that GaussTrace consistently outperforms image-based, geometry-based, and rule-based baselines in 3DGS provenance graph reconstruction.

Our GaussTrace requires no training and assumes no access to editing histories. By generating interpretable explanations for inferred provenance edges, it enables more transparent and explainable provenance analysis for evolving 3D assets. This provides a practical framework for analyzing complex evolutionary relationships among 3DGS models.

## 2. Related work

**3D asset protection.** Recent advances in 3D reconstruction (Kerbl et al., 2023; Lu et al., 2024; Yu et al., 2024; Xie et al., 2024; Lin et al., 2024; Yang et al., 2024; Song et al., 2025; Han et al., 2026; Chen et al., 2024b) have enabled high-fidelity, efficient creation of 3D assets, dramatically expanding their use in gaming, AR/VR, and digital content platforms. Current approaches to protecting these 3D assets primarily rely on watermarking for copyright verification (Luo et al., 2023; Song et al., 2024b; Huang et al., 2024; 2025; Luo et al., 2025; Jang et al., 2025; 2024; Song et al., 2024a; Zhang et al., 2024). For instance, CopyRNeRF (Luo et al., 2023) embeds invisible watermarks into the color representations of NeRF (Mildenhall et al., 2020) while preserving rendering quality. GaussianMarker (Huang et al., 2024) injects imperceptible watermarks via uncertainty-aware perturbations to Gaussian parameters, preserving accuracy under common edits. Geometry Cloak (Song et al., 2024a)

adopts a preventive strategy by embedding imperceptible geometric perturbations into 2D images prior to their use in Triplane Gaussian Splatting (TGS) (Zou et al., 2024), thereby forcing any unauthorized 3D reconstruction to generate a customized, identifiable pattern that functions as a forensic watermark. GS-Hider (Zhang et al., 2024) introduces a steganography framework for 3DGS, replacing spherical harmonic coefficients with secured coupled features that jointly encode the original scene and a hidden message, which are then disentangled by dedicated scene and message decoders for accurate extraction. While existing approaches primarily rely on watermarking for copyright verification, they largely overlook the evolutionary relationships among 3DGS models as they are shared, adapted, and modified across social platforms. This highlights the need for a provenance analysis approach that explicitly captures relationships between different 3DGS models.

**Image provenance analysis.** The study of provenance analysis in the context of online image is pioneered by Kennedy *et al.* (Kennedy & Chang, 2008), who explored inter-image relationships using manipulation-based forensic cues. Subsequently, Dias *et al.* (Dias et al., 2011) formalized a general pipeline that first computes a pairwise dissimilarity matrix and then applies a minimum spanning tree algorithm (Kruskal, 1956) to infer relationships. Bharati *et al.* (Bharati et al., 2017) advanced this direction by estimating dissimilarity through matching of local interest points. However, these early approaches (Bharati et al., 2017; Moreira et al., 2018) heavily rely on handcrafted descriptors such as SURF (Bay et al., 2006) or SIFT (Lowe, 2004), which are sensitive to image manipulations and noise, thereby limiting their robustness and reliability. To mitigate this dependency, Bharati *et al.* (Bharati et al., 2019) proposed an alternative strategy that leverages image metadata, though its effectiveness hinges on the integrity and availability of such auxiliary information. More recently, deep learning has revolutionized provenance analysis by enabling the automatic extraction of discriminative, learnable features. These methods (Zhang et al., 2020; 2025; Bharati et al., 2021) utilize neural representations to construct more accurate provenance graphs. Despite these notable advances in 2D image provenance analysis, existing techniques are not directly transferable to 3DGS models, whose structural and parametric characteristics differ fundamentally from images.

## 3. Preliminary

**3D Gaussian Splatting.** 3DGS (Kerbl et al., 2023) represents 3D scenes as a set of 3D Gaussians to enable efficient and high-quality rendering. Each 3D Gaussian is defined by the function:

$$\mathcal{G}(x) = e^{-\frac{1}{2}(x-\mu)^T \Sigma^{-1}(x-\mu)}, \tag{1}$$

where $x$ denotes an arbitrary point in 3D space, $\mu$ is the mean position of the 3D Gaussian, and $\Sigma$ is its covariance matrix. To ensure that $\Sigma$ remains positive semi-definite, it is constructed from a scaling matrix $S$ and a rotation matrix $R$ as $\Sigma = RSS^T R^T$. In the rendering pipeline, these 3D Gaussians are projected onto the image plane through volume splatting, yielding 2D Gaussians for rasterization. The final pixel color $C$ is computed using a conventional neural point-based strategy (Kopanas et al., 2021; 2022), which composites $N$ depth-ordered Gaussians as:

$$C = \sum_{i \in N} c_i \alpha_i \prod_{j=1}^{i-1}(1 - \alpha_j). \tag{2}$$

Here, $c_i$ is the color estimated from the spherical harmonics (SH) coefficients of the $i$-th Gaussian, and $\alpha_i$ is obtained by evaluating the projected 2D Gaussian with covariance $\Sigma$ (Yifan et al., 2019) and scaling it by a per-point opacity parameter.

Consequently, each 3D Gaussian is fully characterized by five attributes: mean position $\mu$, color $c$, opacity $\alpha$, rotation $r$, and scale $s$, which together constitute the parameter set $\Theta_i = \{\mu_i, c_i, \alpha_i, r_i, s_i\}$.

**Provenance analysis.** Digital provenance analysis (Moreira et al., 2018) is a core task in forensic investigation, aimed at reconstructing the lineage and transformation history among a collection of related digital assets (Bharati et al., 2017; Moreira et al., 2018; Bharati et al., 2019; 2021; Zhang et al., 2025). By inferring how one asset may have been derived from another, this process enables the construction of a provenance graph that reveals the phylogenetic relationships within the set. Formally, let $\mathcal{M} = \{\mathcal{M}_1, \mathcal{M}_2, \ldots, \mathcal{M}_N\}$ denote a set of related digital assets. Provenance analysis proceeds in two stages: 1) a function $\mathcal{D} : \mathcal{M} \times \mathcal{M} \to \mathbb{R}_{\geq 0}$ computes a transformation cost (or dissimilarity) between every pair of models, capturing how likely one could be derived from the other; 2) a graph inference function $\mathcal{R}$ maps the pairwise transformation scores into a directed provenance graph $\mathcal{P} = (\mathcal{V}, \mathcal{E})$, where each node $v_i \in \mathcal{V}$ corresponds to a model $\mathcal{M}_i$, and an edge $(v_i, v_j) \in \mathcal{E}$ indicates that $\mathcal{M}_j$ is a likely descendant of $\mathcal{M}_i$. The resulting graph provides a trace of model evolution and serves as a basis for downstream forensic applications.

## 4. Proposed method

Given a collection of 3DGS models $\{\mathcal{M}_1, \ldots, \mathcal{M}_M\}$, GaussTrace outputs a Directed Acyclic Graph (DAG) where each edge $(\mathcal{M}_i \to \mathcal{M}_j)$ is accompanied by a natural-language rationale explaining the likely transformation. Critically, the pipeline is *training-free*, requires no editing logs, and operates directly on PLY files.

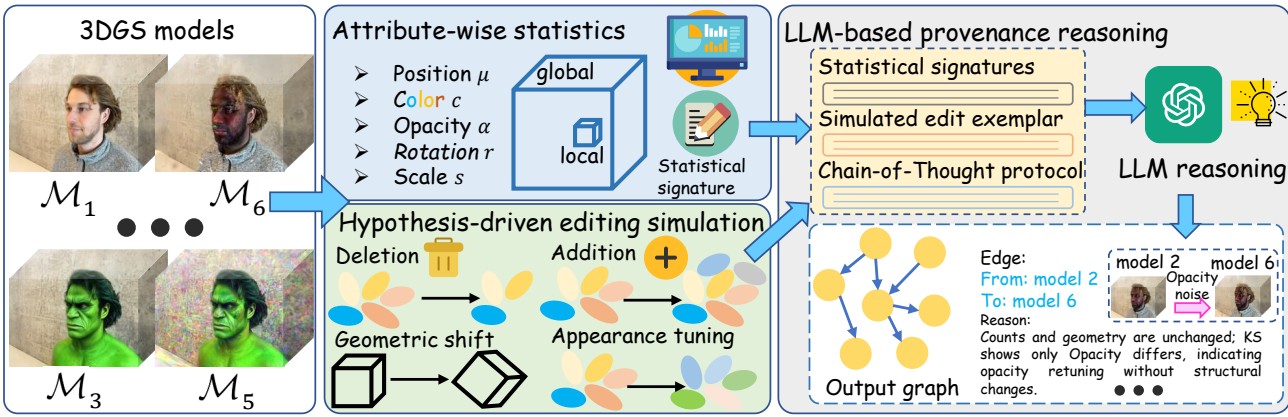

*Figure 2.* Illustration of our GaussTrace framework. First, each 3DGS model is processed by attribute-wise statistical descriptors to capture intrinsic properties. Then, hypothesis-driven editing simulations are performed to generate plausible transformation signatures. These statistical and simulated cues are jointly encoded into a structured prompt. Finally, a Large Language Model (LLM) performs structured Chain-of-Thought reasoning to infer directional provenance relationships and generate explainable edge annotations.

GaussTrace is built upon a dual evidential foundation:

- **Observational evidence**: statistical signatures derived from 3DGS parameter distributions;

- **Hypothesis-driven simulations**: simulated outcomes of plausible editing operations.

These signals are not fused via learned parameters, but are instead analyzed and interpreted through structured Chain-of-Thought reasoning performed by a large language model (LLM). This enables GaussTrace to move beyond similarity-based matching toward *explanatory inference*.

### 4.1. Threat model

We formalize a practical threat model to address the risk of intellectual property infringement and unauthorized manipulation in the context of 3DGS models. The scenario unfolds across three stages involving two principal actors: the model owner and malicious user.

- **Model owner (publication phase)**: The owner creates a 3DGS model and shares it publicly, such as on online platforms or repositories. While the model is intended for legitimate use or further development, the owner expects to retain clear ownership and the ability to track how the model is subsequently used or altered.

- **Malicious user (unauthorized editing and redistribution)**: The malicious user obtains the shared model and modifies it without permission. The altered model is then redistributed, often with no credit to the original creator and sometimes presented as the adversary's own work. This can mislead users and introduce unreliable or harmful content.

- **Model owner (forensics phase)**: The owner can collect suspect 3DGS models from public sources and analyze their internal parameters. Using provenance analysis techniques, the owner can obtain how the model was changed and determine whether it originated from original version, thereby establishing accountability.

### 4.2. Evidential basis for provenance inference

Provenance inference in GaussTrace is grounded in a dual evidential framework that captures both the *observed state* of 3DGS models and the *plausible generative pathways* that could connect them.

**Statistical signature construction.** Each 3DGS model $\mathcal{M}_i$ comprises $N$ 3D Gaussians, parameterized by position $\mu$, color $c$, opacity $\alpha$, rotation $r$ and scale $s$. Rather than treating these as raw feature vectors, we recognize that different attributes respond differently to common edits. For instance, pruning primarily affects count and opacity distribution, while geometric shifts alter position statistics but leave color untouched. To reflect this, we group attributes into five semantic categories: $\mathcal{K} = \{\text{position}, \text{opacity}, \text{scale}, \text{rotation}, \text{color}\}$, and compute statistics per group across all 3D Gaussians.

Formally, the statistical signature of $\mathcal{M}_i$ is defined as

$$\mathbf{s}_i = \mathcal{S}(\mathcal{M}_i), \tag{3}$$

where $\mathcal{S}$ denotes the signature extraction operator, and $\mathbf{s}_i$ is the concatenation of per-group statistics:

$$\mathbf{s}_i = \bigoplus_{k \in \mathcal{K}} \left[ \bar{a}^{(k)}, \sigma^{(k)}, \gamma_1^{(k)}, \gamma_2^{(k)}, H^{(k)} \right]. \tag{4}$$

Here, $\bar{a}^{(k)}$ and $\sigma^{(k)}$ are the mean and standard deviation, $\gamma_1^{(k)}$ and $\gamma_2^{(k)}$ denote skewness and kurtosis, and $H^{(k)}$ is

the entropy of the $k$-th attribute. To better capture localized modifications, we employ multi-scale statistical partitioning. Specifically, the scene bounding box is discretized into a uniform voxel grid, and within each voxel, we compute distributional statistics for Gaussian attributes. For each model, we identify the top-3 voxels exhibiting the greatest statistical divergence and incorporate them with the global statistics into the LLM prompt. This enables the LLM to perform spatially grounded, region-aware reasoning about localized edits.

For any unordered pair $(\mathcal{M}_i, \mathcal{M}_j)$, we assess distributional divergence via two-sample Kolmogorov–Smirnov (KS) tests on each attribute group $k$, yielding $p$-values $p_{ij}^{(k)}$. Attributes with $p_{ij}^{(k)} < 0.05$ are flagged as significantly altered. Crucially, we do not rely on raw effect sizes (*e.g.*, absolute parameter differences); instead, we prioritize statistical significance, as even subtle but systematic changes (such as uniform opacity scaling) can strongly indicate intentional editing.

**Hypothesis-driven editing simulation.** While statistical signatures reveal *what* models differ, they do not explain *why*. To bridge this gap, we introduce a library of hypothesis-driven simulations that encode common user behaviors in 3DGS workflows. Our design is guided by empirical observation of editing patterns in public 3DGS repositories and community tutorials, which reveal four dominant operations:

1. **Deletion**: removal of low-opacity or redundant Gaussians to reduce render cost;

2. **Addition**: addition of Gaussians in under-represented regions to improve fidelity;

3. **Geometric shift**: application of geometric transformations (*e.g.*, translation and rotation) to better align the model with external reference data;

4. **Appearance tuning**: color or opacity adjustments for stylistic or lighting correction.

Each editing operation $o \in \mathcal{O} = \{o_1, \ldots, o_4\}$ defines a deterministic transformation $\mathcal{T}_o : \mathcal{M}_i \mapsto \mathcal{M}_i^o$. For deletion, we remove the 20% lowest-opacity Gaussians; for addition, we sample new Gaussians whose parameters follow the empirical distributions; geometric shift applies random translation and rotation; appearance tuning adds Gaussian noise to color coefficients.

The impact of each edit is quantified via the Average Treatment Effect (ATE) style score on a set of key metrics $\mathcal{Q}$:

$$\text{ATE}_q(o; \mathcal{M}_i) = q\big(\mathcal{S}(\mathcal{M}_i^o)\big) - q\big(\mathcal{S}(\mathcal{M}_i)\big), \quad \forall q \in \mathcal{Q}. \quad (5)$$

These ATEs are not used as features in a classifier, but as *semantic anchors* in the reasoning prompt. For example, if the observed Gaussian count decreases by 15% while positional and color statistics remain largely unchanged, and pruning in our simulation library produces a similar pattern of changes, the LLM can infer pruning as a plausible cause, even if the actual pruning ratio differs. This makes the framework robust to variations in editing intensity.

The four simulated operations are representative examples of common editing behaviors, not an exhaustive taxonomy. *Our reasoning framework **does not restrict the LLM to these categories alone**. It is encouraged to hypothesize other plausible transformations when the statistical evidence suggests edits beyond the simulated set. This design ensures flexibility and adaptability to unseen or composite editing behaviors.* Moreover, each simulations are performed *from both directions (i.e., $\mathcal{M}_i \to \mathcal{M}_i^o$ and $\mathcal{M}_j \to \mathcal{M}_j^o$)* to support directionality disambiguation during reasoning.

### 4.3. LLM-based provenance reasoning

The final inference stage transcends pattern matching by treating provenance as a *structured reasoning problem*. A large language model (LLM) is employed not as a black-box predictor, but as a symbolic engine that synthesizes heterogeneous evidence into a coherent causal narrative.

The input to the LLM is a carefully curated prompt that encodes three types of information. An illustrative excerpt of the prompt is shown below:

> You are an expert in 3D Gaussian Splatting (3DGS) and causal inference. Build a provenance graph where nodes are models and edges represent transformations.
> Input: **Statistical signatures**: per-model distributional statistics (e.g., count, entropy, variance) for position, color, opacity, scale, and rotation;
> **Simulated edit exemplars**: illustrative transformations (deletion, addition, geometric shift, appearance tuning) with their ATE;
> **Pairwise comparisons**: KS test $p$-values indicating statistically significant differences between model pairs.

The prompt enforces a four-step Chain-of-Thought (CoT) protocol:

> Reasoning Protocol: The LLM performs inference by (i) treating statistical signatures as primary evidence, (ii) using simulated edits as semantic anchors (not constraints), (iii) confirming changes via significant KS tests ($p < 0.05$), and (iv) hypothesizing transformations flexibly. Output: a JSON-formatted provenance graph with human-readable edge justifications.

*The complete prompt template is provided in the appendix.*

The output is a natural-language justification that explicitly links evidence to conclusion. This design ensures that every provenance edge is not only accurate but also *auditable*: a human expert can verify the reasoning by inspecting the cited statistics and simulations. By grounding reasoning in the extracted evidence, GaussTrace establishes a new paradigm for transparent, explainable, and forensically viable 3D asset provenance.

### 4.4. Implementation details

Our GaussTrace is implemented in Python 3.8. The experiments are conducted on a single workstation equipped with an Intel Core i7-14700 CPU and an NVIDIA RTX 4090 GPU. For each 3DGS model, we extract core attributes from the PLY file, including 3D positions, colors, opacities, scales, and rotations. To capture localized edits, we perform multi-scale analysis by partitioning the scene bounding box into a $4 \times 4 \times 4$ voxel grid. Voxels containing fewer than 10 Gaussians are excluded to ensure statistical reliability. From the remaining voxels, we select the top-3 exhibiting the largest change and include their statistics in the LLM prompt. For each model, we perform four common editing operations as described in Section 4.2. The resulting edited models, along with their associated ATEs, are incorporated into the reasoning prompt. We employ `GPT-5` as the reasoning engine to perform provenance analysis. The model receives structured evidence and follows a chain-of-thought protocol to output a provenance graph in JSON format.

## 5. Experiments

### 5.1. Experimental settings

**Dataset.** To evaluate our proposed method, we create a comprehensive 3DGS provenance graph dataset featuring diverse scenes, each paired with accurate ground-truth labels. We build this dataset by modifying 3DGS models from the Mip-NeRF360 (Barron et al., 2022), Instruct-NeRF2NeRF (Haque et al., 2023), and Deep Blending (Hedman et al., 2018) datasets. These modifications include applying the 3DGS editing technique (Chen et al., 2024a), merging distinct 3DGS models, executing geometric transformations like translations and rotations, and introducing noise to the models. All original 3DGS models are trained using standard configurations (Kerbl et al., 2023). In total, the dataset includes 44 3DGS models, encompassing a wide variety of scene types to ensure comprehensive evaluation, consistent with the scale of scenes used in current 3DGS editing and forensic studies (Chen et al., 2024a; Meng et al., 2025). *More details are provided in the appendix.*

**Baselines.** To the best of our knowledge, there is no existing method specifically designed for 3DGS models. Therefore, we compare GaussTrace against a comprehensive set

*Table 1.* Quantitative evaluation of provenance graph construction for 3DGS models. Reported values are the mean of three standard metrics across multiple scenes, *i.e.*, Vertex Overlap (VO), Edge Overlap (EO), and Vertex Edge Overlap (VEO). Points CD denotes Chamfer Distance computed on Gaussian centers. Higher values indicate better performance (↑). Best results are in **bold**.

| Method | VO↑ | EO↑ | VEO↑ |
|---|---|---|---|
| ResNet+MI | 1.000 | 0.205 | 0.627 |
| DenseNet+MI | 1.000 | 0.212 | 0.629 |
| ViT-B+MI | 1.000 | 0.230 | 0.639 |
| Points CD+MI | 0.758 | 0.268 | 0.531 |
| ResNet+Integrity | 1.000 | 0.297 | 0.669 |
| DenseNet+Integrity | 1.000 | 0.319 | 0.680 |
| ViT-B+Integrity | 1.000 | 0.470 | 0.751 |
| Points CD+Integrity | 0.758 | 0.270 | 0.532 |
| Rule-based search | 0.939 | 0.319 | 0.566 |
| GaussTrace w/o CoT | 1.000 | 0.811 | 0.913 |
| Proposed GaussTrace | **1.000** | **0.890** | **0.948** |

of baseline strategies that combine established strategies for link prediction (whether two models are related) and direction prediction (which is the ancestor). For link prediction, we consider two families of approaches: 1) **Image-based method**: we extract features of rendered images using ResNet (He et al., 2016), DenseNet (Huang et al., 2017), and ViT-B (Dosovitskiy et al., 2020). The $L_2$ distance between feature vectors is used as the dissimilarity measure. 2) **Geometry-based method**: considering the point cloud-like nature of 3DGS, we compute the **Chamfer Distance** (Butt & Maragos, 1998) between the sets of Gaussian centers as dissimilarity metric. For direction prediction, we adopt two established forensic strategies: 1) **Mutual information (MI)-based method** (Moreira et al., 2018): following (Moreira et al., 2018), we estimate the MI-based dissimilarity between rendered image pairs and assume the model with higher value is the ancestor. 2) **Integrity score-based method** (Zhang et al., 2020): as in (Zhang et al., 2020), we compute an integrity score for each image (using the same publicly released scorer (Guillaro et al., 2023) as in (Zhang et al., 2025)) and treat the model with higher integrity as the ancestor.

In addition, we introduce a **rule-based search** (RBS) baseline that directly operates on 3DGS parameters. RBS enumerates plausible editing operations, simulates their effects on candidate source models, and selects the transformation that best explains the observed statistical differences via a hand-crafted scoring function. RBS leverages the native parameter space of 3DGS and incorporates domain knowledge about common editing workflows. *More details are provided in the appendix.*

**Evaluation methodology.** We evaluate the proposed approach using standard metrics for provenance graph construction (Bharati et al., 2021; Moreira et al., 2018; Zhang et al., 2025), which measure the alignment between the generated graph and the ground-truth graph. The assessment employs three F1 score-based metrics, *i.e.,* Vertex Overlap (VO) for node correctness, Edge Overlap (EO) for relational accuracy, and Vertex Edge Overlap (VEO) for an overall graph-level match that jointly considers nodes and edges. Their precise formulations are provided in the appendix. To further validate robustness, we test our approach under challenging conditions, such as the inclusion of irrelevant distractor models.

### 5.2. Experimental results

**Quantitative results.** We compare GaussTrace against a range of baseline methods for provenance graph construction on 3DGS models. Specifically, we adapt image-based approaches by combining standard feature extractors, *i.e.,* ResNet (He et al., 2016), DenseNet (Huang et al., 2017), ViT-B (Dosovitskiy et al., 2020), and points-based Chamfer Distance (Points CD) (Butt & Maragos, 1998) with two representative direction prediction methods: mutual information (MI)-based method (Moreira et al., 2018) and Integrity score-based method (Zhang et al., 2020). All methods are evaluated using the standard metrics VO, EO, and VEO, averaged over multiple scenes. As shown in Table 1, these baseline methods consistently struggle to infer the correct transformation relationships between 3DGS models, as evidenced by low EO scores, ranging from 0.205 to 0.470, and corresponding VEO values below 0.76. This suggests that naive adaptation of 2D image-based features or geometric distances to the 3DGS provenance setting is insufficient for capturing the trace of 3D manipulation operations.

We further introduce a rule-based search (RBS) baseline that operates directly on 3DGS parameters and leverages the same statistical descriptors and editing simulations as GaussTrace, but replaces LLM-based reasoning with a deterministic scoring mechanism. Despite using domain-specific evidence, RBS achieves only EO = 0.319 and VEO = 0.566, demonstrating that simple matching of simulated and observed statistics is inadequate for inferring directional provenance relationships. In contrast, GaussTrace achieves VO = 1.000, EO = 0.890, and VEO = 0.948, significantly outperforming all baselines. These results highlight that accurate 3DGS provenance analysis requires not only informative evidence extraction, but also structured reasoning, which is effectively supported by our LLM-based framework.

**Qualitative results.** To visually assess the structural and directional accuracy of provenance analysis, we compare the constructed graphs from our method against two representative baseline approaches: ViT-B (Dosovitskiy et al.,

*Table 2.* Robustness of GaussTrace to distractor models in provenance graph construction. We evaluate performance under increasing distractor ratios (0%, 10%, 20%) using standard metrics, *i.e.,* Vertex Overlap (VO), Edge Overlap (EO), and Vertex Edge Overlap (VEO). Higher values indicate better performance (↑).

| Setting | Distractor ratio | VO↑ | EO↑ | VEO↑ |
|---------|------------------|-------|-------|-------|
| (a) | 0% | 1.000 | 0.890 | 0.948 |
| (b) | 10% | 0.969 | 0.811 | 0.894 |
| (c) | 20% | 0.940 | 0.795 | 0.871 |

2020) + Integrity (Zhang et al., 2020) and Points Chamfer Distance (CD) (Butt & Maragos, 1998) + Integrity (Zhang et al., 2020). As shown in Figure 3, the ground-truth graph is compared with predictions in terms of both edge correctness and directionality. Our method recovers the full evolutionary structure with high fidelity, which produces green edges (correct connections with correct directions) that closely match the reference. In contrast, the baselines exhibit numerous pink edges (correct connections but incorrect directions) and red edges (wrong connections), indicating difficulties in predicting accurate relationships. This visual comparison highlights the limitations of adapting 2D image features or geometric distances. The results demonstrate that our GaussTrace not only identifies plausible links but also infers their direction accurately, enabling reliable reconstruction of model evolution.

**Robustness evaluation.** To assess the robustness of GaussTrace, we evaluate its performance in the presence of distractor models, *i.e.,* unrelated 3DGS models that share no derivation relationship with the true set. We construct three test settings by injecting distractors at ratios of 0%, 10%, and 20% into the input collection and measure provenance graph accuracy using VO, EO, and VEO. As shown in Table 2, GaussTrace achieves near-perfect vertex recovery (VO = 1.000) in the clean setting and remains highly accurate even with 20% distractors (VO = 0.940). Although EO and VEO decrease slightly with more distractors, the consistently high scores confirm GaussTrace's robustness to irrelevant content, demonstrating its reliability in noisy or imperfect real-world settings.

**Effectiveness of CoT.** To evaluate the effectiveness of structured Chain-of-Thought (CoT) reasoning, we compare the full GaussTrace framework with a variant that bypasses the CoT protocol (denoted *GaussTrace w/o CoT*). In the ablated version, the large language model directly predicts edges from statistical and simulated evidence without explicit step-by-step inference. As shown in Table 1, removing the CoT component leads to a noticeable drop in edge-level accuracy, *i.e.,* EO decreases from 0.890 to 0.811, and VEO drops from 0.948 to 0.913, while vertex identification (VO) remains perfect. This demonstrates that CoT reasoning

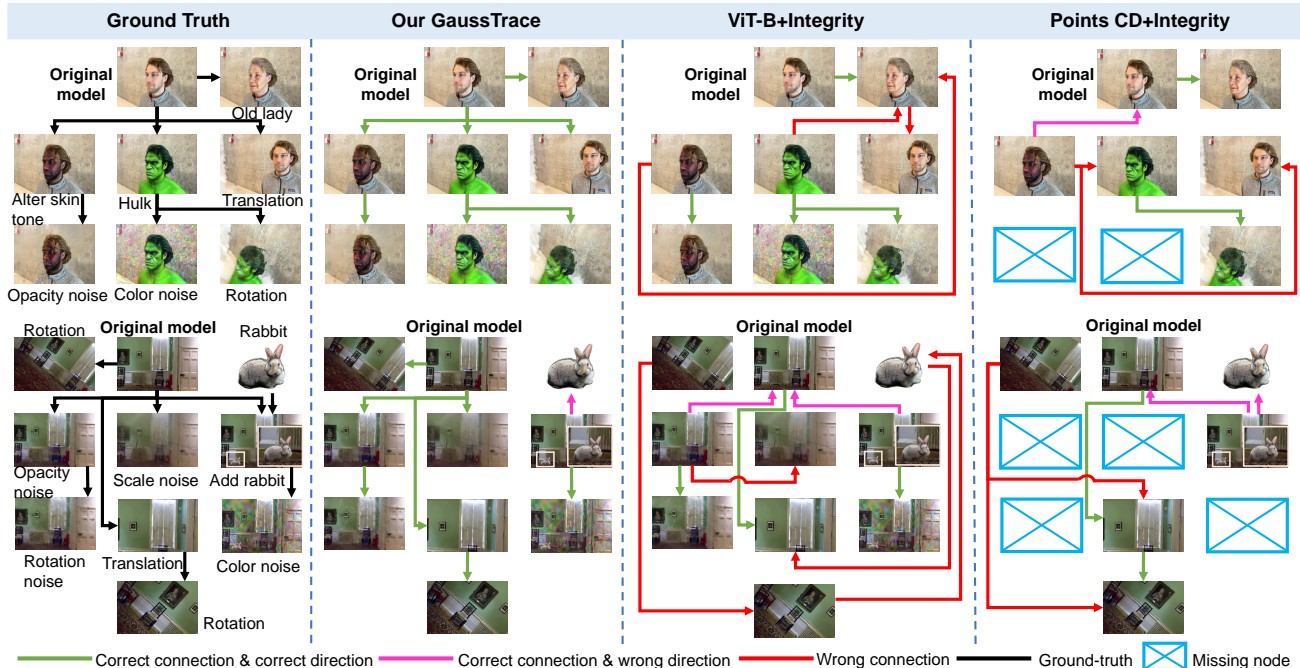

*Figure 3.* Qualitative comparison of 3DGS provenance graph construction. From left to right: ground-truth, Our GaussTrace, ViT-B (Dosovitskiy et al., 2020)+Integrity (Zhang et al., 2020) and Points CD (Butt & Maragos, 1998)+Integrity (Zhang et al., 2020). The green edges indicate correct connections with correct directions, pink edges denote correct connections with wrong directions, and red edges represent wrong connections. Missing node represents that this node is absent from the provenance graph constructed by this method. Our method can construct the evolutionary relationships among 3DGS models accurately, while baseline methods struggle with directionality and false links. For visualization clarity, we display a rendered view of each model, while our analysis is performed on the 3DGS models. **Please zoom in for better view.**

plays a critical role in interpreting complex statistical patterns and translating low-level statistical and simulated cues into accurate transformation relationship. By guiding the model through a principled reasoning process, CoT enables more accurate and reliable provenance graph construction, particularly for edge prediction, which is important to capture evolutionary relationships of different 3DGS models.

**Different 3DGS attributes.** To assess the impact of individual 3D Gaussian attributes in provenance analysis, we conduct experiments by excluding each attribute group (*i.e.*, position $\mu$, opacity $\alpha$, scale $s$, rotation $r$, and color $c$) from the statistical signature while keeping all others. As shown in Figure 4, removing any individual attribute leads to a measurable drop in edge-level accuracy, confirming that all attributes contribute meaningfully to provenance analysis. Although VO remains at 1.000 in all cases, indicating robust node identification, the consistent decline in EO and VEO demonstrates that no attribute is redundant. Each captures distinct aspects of editing behavior, *e.g.*, geometric structure and appearance. This validates our design choice to perform comprehensive, multi-attribute statistical profiling, as partial signatures inevitably reduce forensic discriminability and limit the reliability of transformation reasoning.

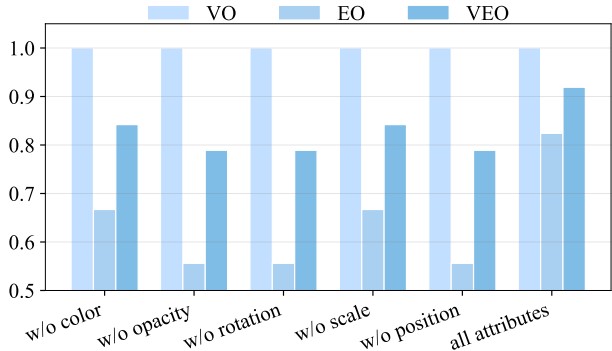

*Figure 4.* Impact of individual 3D Gaussian attributes on provenance graph construction performance of GaussTrace. The Gaussian attributes include position ($\mu$), opacity ($\alpha$), scale ($s$), rotation ($r$), and color ($c$).

**Explainability.** Our method can generate explanations for each directed edge in the constructed 3DGS provenance graph. As illustrated in Figure 5, each directed relationship is accompanied by a natural-language rationale that explains the likely editing operation and the statistical evidence supporting it. Notably, our approach handles composite edits, where multiple transformations are applied jointly. For in-

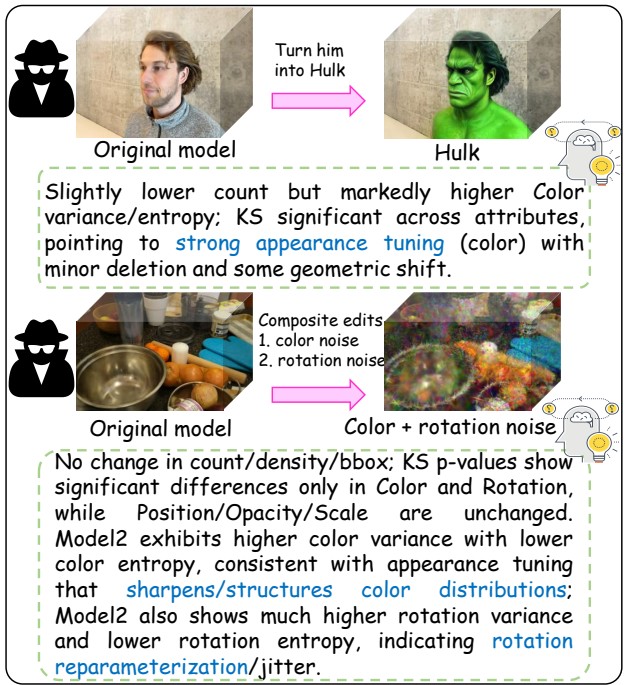

Figure 5. Illustration of explanations for provenance edges. Our method can generate human-readable justifications for each directed relationship in the provenance graph, enabling traceable and interpretable model evolution.

stance, in the example shown in the Figure 5, one model is derived from another through a combination of color adjustment and rotation noise. GaussTrace correctly identifies this complex transformation and articulates the joint effect in its explanation. By grounding each inference in interpretable cues, our method enables users to audit and verify provenance claims. This turns the provenance graph from a black-box prediction into a transparent, traceable record of model evolution.

## 6. Conclusion

In this paper, we present GaussTrace, a novel framework for constructing directed provenance graphs of 3DGS models. Our approach addresses the challenge of tracing evolutionary relationships in shared and modified 3D assets through an evidence-based reasoning approach. Specifically, we build evidential inputs by combining attribute-wise statistical profiling of 3DGS parameters with hypothesis-driven simulations of common editing operations. By integrating these cues into a structured Chain-of-Thought prompt for large language model, GaussTrace enables directional provenance inference with natural-language explanations. Extensive experimental results show that our method achieves accurate provenance graph construction, making it effective for protection and forensic tracing of 3DGS assets.

Future work will further explore more scalable and efficient paradigms for large-scale 3D provenance analysis, particularly for increasingly complex multi-source editing and redistribution scenarios. We are also interested in investigating whether emerging computational paradigms, such as quantum machine learning, may provide new opportunities for accelerating high-dimensional statistical analysis and complex provenance graph reasoning in large-scale 3DGS asset evolution.

## Acknowledgements

This work was carried out at the Renjie Group, Hong Kong Baptist University. Renjie Group is supported by the National Natural Science Foundation of China under Grant No. 62302415, Guangdong Basic and Applied Basic Research Foundation under Grant No. 2024A1515012822, and the Research Grant Council of Hong Kong SAR, under a GRF Grant 12203124 and an ECS Grant 22201125. This work was supported by the Beijing Municipal Science & Technology Program No. Z251100007125021. Ziyuan Luo is supported by a fellowship award from the RGC of Hong Kong SAR (Project No. HKBU JRFS2627-2S03).

## Impact Statement

This work aims to advance trustworthy and accountable generative 3D ecosystems by enabling forensic traceability of 3DGS models. By reconstructing explainable directed provenance graphs without requiring training or edit logs, GaussTrace can help protect creators' intellectual property, deter unauthorized redistribution, and support digital forensics in cases of malicious content propagation.

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

## A. Overview

This appendix provides more explanations, implementation details, and additional experimental results to accompany the main paper. The contents are organized as follows:

- Section B provides implementation details of the baseline methods.

- Section C gives the detailed description of our 3DGS provenance graph dataset.

- Section D presents formal definitions of the evaluation metrics.

- Section E presents an analysis of GaussTrace's performance across different large language models.

- Section F is a discussion of current limitations and future work.

- Section G shows additional qualitative results across diverse 3D scenes.

- Section H provides the LLM prompt template used in our framework.

## B. Implementation details of baseline methods

Since no existing method is specifically designed for 3DGS provenance analysis, we construct baseline approaches by combining standard strategies for **link prediction** (assessing whether two models are related) and **direction prediction** (orienting edges to identify ancestors). The full pipeline is as follows.

**Link prediction.** For a set of $N$ 3DGS models, we consider two families of dissimilarity measures:

1. **Image-based method**: we extract features of rendered images using ResNet-18 (He et al., 2016), DenseNet-121 (Huang et al., 2017), and ViT-B/16 (Dosovitskiy et al., 2020). The pairwise $L_2$ distance between image patch features is computed to form a symmetric dissimilarity matrix $\mathbf{D} \in \mathbb{R}^{N \times N}$. This approach adheres to the strategy used in TAE (Bharati et al., 2021).

2. **Geometry-based method**: we treat Gaussian centers $\{\mu_i\}$ as a point cloud and compute the symmetric Chamfer Distance (CD) (Butt & Maragos, 1998) between all model pairs, yielding another dissimilarity matrix $\mathbf{D}$.

Given $\mathbf{D}$, we construct an undirected provenance graph by applying the Minimum Spanning Tree (MST) (Kruskal, 1956) algorithm. The MST connects different nodes with minimal total dissimilarity, producing a tree structure that serves as the link prediction output, which is consistent with prior image provenance analysis works (Bharati et al., 2017; 2021).

**Direction prediction.** Once the undirected graph is obtained, we assign directions to edges using one of two established forensic strategies:

1. **Mutual Information (MI)** (Moreira et al., 2018): for each edge $(i, j)$, we estimate the MI-based dissimilarity between the rendered images of models $i$ and $j$ using the strategy in (Moreira et al., 2018). The node with higher value is designated as the ancestor (parent).

2. **Integrity score** (Zhang et al., 2020): using the public TruFor scorer (Guillaro et al., 2023) (as in (Zhang et al., 2025)), we compute an integrity score for each rendered image. The model with the higher integrity is assigned as the ancestor.

**Baseline configurations.** Combining the four dissimilarity measures (ResNet (He et al., 2016), DenseNet (Huang et al., 2017), ViT-B (Dosovitskiy et al., 2020), Points CD (Butt & Maragos, 1998)) with the two direction assignment strategies (MI (Moreira et al., 2018), Integrity (Zhang et al., 2020)) yields the following eight baselines: ResNet+MI, ResNet+Integrity, DenseNet+MI, DenseNet+Integrity, ViT-B+MI, ViT-B+Integrity, Points CD+MI, and Points CD+Integrity.

**Rule-based search.** To assess whether the large language model in GaussTrace provides genuine reasoning advantages over direct statistical matching, we introduce a **Rule-Based Search** (RBS) baseline that operates solely on 3DGS parameters without learning or linguistic reasoning. It systematically tests candidate editing operations and selects the one that best explains the observed differences between two 3DGS models.

Given a pair of 3DGS models $(A, B)$, RBS performs the following steps:

1. compute statistical profiles: extract the same set of statistical descriptors used in GaussTrace (*e.g.*, count, density, entropy of position/opacity/color, etc.) for both $A$ and $B$.

2. enumerate candidate edits: consider four common editing operations: deletion, addition, geometric shift and appearance tuning.

3. simulate transformations: for each candidate operation $o$, apply our deterministic simulator to $A$ to obtain a simulated model $A_o$, then compute its statistical profile.

4. compute the inconsistency score:

$$\text{score}(o) = \frac{1}{|\mathcal{F}|} \sum_{f \in \mathcal{F}} |f(A_o) - f(B)|, \tag{6}$$

where $\mathcal{F}$ is the set of statistical features (*e.g.*, count, opacity_mean, pos_entropy).

RBS selects the operation with the minimal score as the predicted transformation. To infer directionality, it compares the best scores for $A \rightarrow B$ and $B \rightarrow A$, and assigns the edge in the direction with the lower score.

Critically, to avoid spurious edges between unrelated models, we apply a threshold: an edge is only added if the minimal score is below the threshold. This ensures that only statistically plausible relationships are retained, mimicking a conservative forensic stance.

This design guarantees a fair comparison: RBS uses the exact same statistical descriptors and simulation modules as GaussTrace, the only difference lies in the inference mechanism (rule-based scoring vs. LLM-guided reasoning).

All methods produce a directed provenance graph, and their performance is evaluated against the ground-truth directed provenance graph using VO, EO, and VEO.

## C. Dataset description

To enable evaluation of provenance analysis in realistic 3D manipulation workflows, we construct a diverse and semantically rich dataset of 3DGS models, grounded in several representative scenes from established 3D reconstruction benchmarks: bonsai and counter from Mip-NeRF360 (Barron et al., 2022), face from InstructNeRF2NeRF (Haque et al., 2023), and playroom and drjohnson from Deep Blending (Hedman et al., 2018) dataset. Each scene is first trained into a high-fidelity 3DGS model using standard training configurations (Kerbl et al., 2023), serving as the root of its provenance graph.

We then generate multiple derived variants by applying real-world editing operations via the 3DGS editing framework (Chen et al., 2024a), which enables intuitive, user-guided manipulation of 3DGS. In the face scene, for instance, we transform the original model by altering its appearance (*e.g.*, turn him into hulk), effectively simulating a semantic recoloring like real-world editing. In the playroom scene, we introduce entirely new objects, such as a teddybear, by synthesizing and inserting new Gaussians into empty regions, mimicking how users might augment a scene with personal items.

Beyond semantic edits, we also perform geometric transformations, including translations and rotations of entire object regions. Noise is also introduced to different 3D Gaussian attributes (*e.g.*, scaling opacity across the model), capturing the degradation and refinement patterns common in iterative editing workflows.

These operations are applied in varying combinations, yielding a total of 44 distinct 3DGS models. Every edit is logged to construct a ground-truth directed graph, where nodes correspond to models and edges represent the transformation. The resulting dataset captures a wide range of realistic editing behaviors, from semantic alterations to structural modifications, making it suited for evaluating 3DGS provenance analysis methods.

## D. Formulations of evaluation metrics

We evaluate the quality of a predicted provenance graph $\mathcal{P}' = (\mathcal{V}', \mathcal{E}')$ against the ground-truth provenance graph $\mathcal{P} = (\mathcal{V}, \mathcal{E})$, where each node $v_i \in \mathcal{V}$ (or $\mathcal{V}'$) corresponds to a 3DGS model $\mathcal{M}_i$, and a directed edge $(v_i, v_j) \in \mathcal{E}$ indicates that $\mathcal{M}_j$ is a descendant of $\mathcal{M}_i$.

Following established practice in digital provenance analysis (Bharati et al., 2021; Moreira et al., 2018; Zhang et al., 2025), we adopt three F1-based metrics, *i.e.*, Vertex Overlap (VO) measures node-level accuracy:

$$VO(\mathcal{P}, \mathcal{P}') = 2 \times \frac{|\mathcal{V}' \cap \mathcal{V}|}{|\mathcal{V}'| + |\mathcal{V}|}, \tag{7}$$

Edge Overlap (EO) evaluates the correctness of directional relationships:

$$EO(\mathcal{P}, \mathcal{P}') = 2 \times \frac{|\mathcal{E}' \cap \mathcal{E}|}{|\mathcal{E}'| + |\mathcal{E}|}, \tag{8}$$

Vertex Edge Overlap (VEO) provides a unified score combining both nodes and edges:

$$VEO(\mathcal{P}, \mathcal{P}') = 2 \times \frac{|\mathcal{V}' \cap \mathcal{V}| + |\mathcal{E}' \cap \mathcal{E}|}{|\mathcal{V}'| + |\mathcal{V}| + |\mathcal{E}'| + |\mathcal{E}|}, \tag{9}$$

these metrics compute the F1 score between the predicted and ground-truth graph. For experiments involving multiple scenes, we compute the metric values for each scene individually and report the average across all scenes as the final result.

## E. Robustness to LLM choice

To evaluate the dependency of GaussTrace on specific large language models, we test our framework with four LLMs, *i.e.*, GPT-5, Qwen3-Max, Llama-4, and Gemini-2.5-Pro, on the *counter* scenario. As shown in Table 3, all models achieve perfect node matching (VO = 1.0), confirming that our evidence encoding is universally interpretable. While minor variations appear in Edge Overlap (EO) and Vertex Edge Overlap (VEO), the overall performance remains high across all models. This demonstrates that GaussTrace's effectiveness stems from its structured evidence design rather than reliance on a particular LLM, supporting its practical deployability in diverse environments.

*Table 3.* Provenance inference performance of GaussTrace using different large language models (LLMs) on the *counter* scenario. All models achieve excellent performance, demonstrating the robustness of our framework across LLMs.

| LLM | VO ↑ | EO ↑ | VEO ↑ |
|---|---|---|---|
| GPT-5 | 1.000 | 1.000 | 1.000 |
| Qwen3-Max | 1.000 | 0.857 | 0.933 |
| Llama-4 | 1.000 | 0.933 | 0.968 |
| Gemini-2.5-Pro | 1.000 | 1.000 | 1.000 |

## F. Limitations and future work

While GaussTrace demonstrates strong performance in constructing provenance graphs for 3DGS assets, several limitations remain. First, the current benchmark is still relatively small and controlled compared to fully unconstrained real-world 3DGS redistribution pipelines. Although the dataset is constructed using realistic editing operations and diverse transformation chains, real-world scenarios may involve more heterogeneous editing tools, compression artifacts, and multi-party modifications. Expanding toward larger and more realistic provenance benchmarks therefore remains an important direction for future work.

Second, the current framework involves pairwise comparison and graph-level reasoning, which may introduce scalability challenges for very large model collections. While the current graph sizes remain computationally manageable in practice and lightweight pre-filtering can reduce the number of candidate comparisons, improving computational efficiency and reducing LLM inference cost remain important future directions.

Finally, GaussTrace remains a technical solution. Our method focuses on algorithmic inference of evolutionary relationships but does not, by itself, enforce intellectual property rights or prevent malicious tampering. Provenance, in this context, is a diagnostic signal rather than a protective barrier. For such signals to translate into real-world accountability, they must be integrated into a broader ecosystem, including standardized metadata formats, platform-level verification pipelines, and legal frameworks that recognize machine-inferred relationships as credible evidence. Future work should therefore explore how to co-design technical provenance systems with governance structures, ensuring that algorithmic insights can be trusted, verified, and acted upon across the 3D content pipeline.

## G. Additional qualitative results

This section presents additional qualitative comparisons of provenance graph reconstruction across a diverse set of 3D scenes, complementing the qualitative analysis in the main paper. As shown in Figure 7 to Figure 8, GaussTrace consistently recovers the correct topology and direction of transformation edges across complex editing scenarios. Crucially, GaussTrace does not rely on rigid template matching. Instead, it infers transformations by grounding reasoning in statistical evidence across both global and local scales. This enables the framework to reconstruct not only the presence of edits, but also their likely causes. As shown in Figure 6, GaussTrace can further enrich the provenance graph with semantic context and support more informed interpretation of 3DGS model evolution.

## H. Prompt template

We provide the full prompt template used to guide the Large Language Model (LLM) in provenance reasoning. The prompt includes several key components: statistical signatures of input 3DGS models, illustrative examples of simulated editing operations with their Average Treatment Effects (ATEs), and pairwise statistical comparison results. Below, we present a representative instance of the prompt structure to illustrate its format and content organization.

---

**Evidential basis prompt**

You are an expert in 3D Gaussian Splatting (3DGS) and causal inference. Build a provenance graph where nodes are models and edges represent transformations.
Input:
%Global Statistical Signatures
- Model id: [model_count] Gaussians
- Density: [density]
- Bounding Box Volume: [bbox_volume]
- Position: [variance], [skew], [kurtosis], [entropy]
- Color: [mean], [variance], [skew], [kurtosis], [entropy]
- Opacity: [mean], [skew], [kurtosis], [entropy]
- Scale: [mean], [variance], [skew], [kurtosis], [entropy]
- Rotation: [mean], [variance], [skew], [kurtosis], [entropy]
%Local Regions
Model [id] — region [id] ([region_ratio]): [count], [pos entropy], [color entropy]
%Simulated Transformations (Examples)
- Model [id] Deletion (example):
[count], [pos_entropy], [ATE_entropy]
- Model [id] Addition (example):
[count], [pos_entropy], [ATE_entropy]
- Model [id] Geometric Shift (example):
[pos_var], [ATE_pos_var]
- Model [id] Appearance Tuning (example):
[color entropy], [ATE_color_entropy]
%Pairwise Statistical Comparisons
- Model [i] vs. Model [j]:
- KS p-values: [position], [color], [opacity], [scale], [rotation]

---

It also includes explicit instructions for chain-of-thought reasoning and structured JSON output. Below, we provide the detailed reasoning protocol to clarify how the LLM integrates statistical evidence and simulation cues to construct evolutionary relationships among diverse models.

---

**Reasoning protocol**

Task (Chain-of-Thought with Causal Inference):

Step 1: Compare stats: Lower count/higher entropy may indicate simplification; higher count/variance may indicate added details. Use all stats as primary evidence.

Step 2: Analyze simulated transformations as examples only: These are illustrative of common edits (e.g., deletion/addition); if they match, use them to support inference, but do not rely solely on them if stats suggest other transformations.

Step 3: Use KS p-values ($<0.05$ = significant) to confirm changes, considering simulations as auxiliary.

Step 4: Hypothesize transformations (e.g., addition, deletion, geometric shift, appearance tuning, or others inferred from stats), allowing flexibility beyond provided examples.

Step 5: Output JSON:

"nodes": ["id": "Model1", "stats": "count": ..., "density": ..., ..., ...],

"edges": ["from": "Model1", "to": "Model2", "reason": "Model 2 has lower count but higher entropy indicating simplification"]

Do not include additional text.

---

The complete prompt is conditioned on the input models during inference and enables the LLM to generate structured, explainable provenance graphs of 3DGS models.

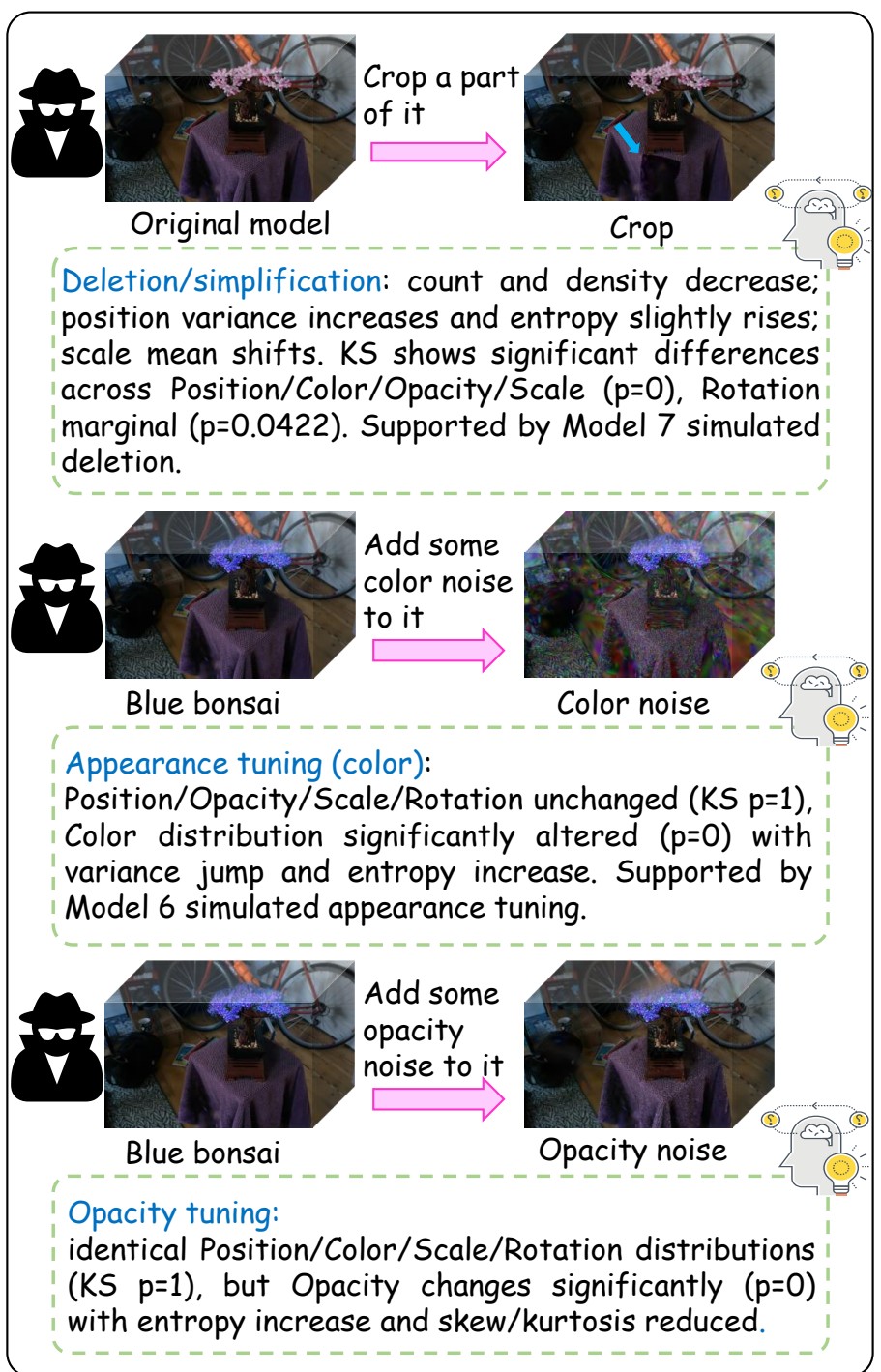

*Figure 6.* Example edge explanations produced by our method. Each directed relationship in the provenance graph is accompanied by a natural-language rationale that clarifies the underlying transformation, supporting transparent model evolution.

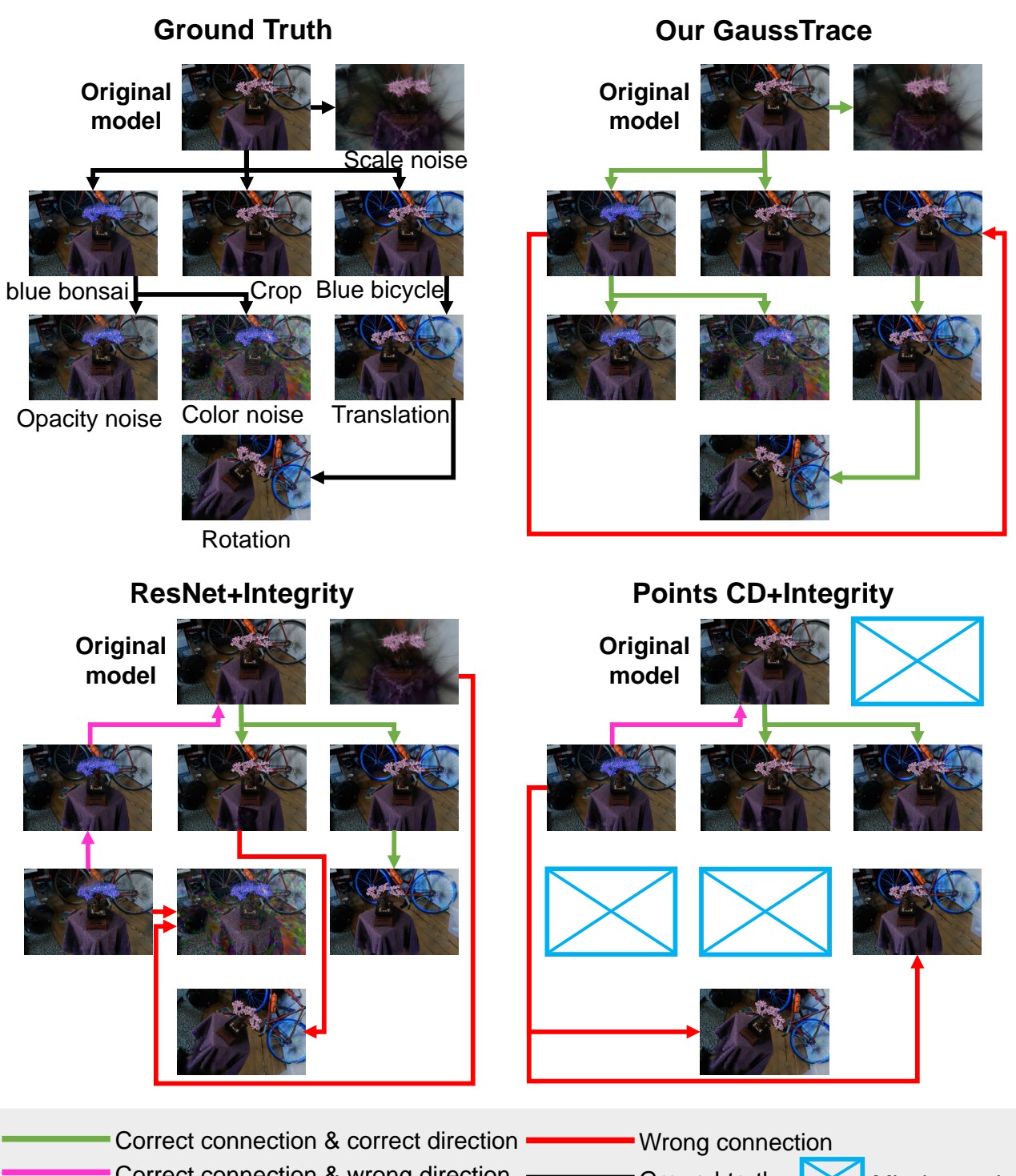

*Figure 7.* Qualitative comparison of 3DGS provenance graph construction. Top row, left to right: ground-truth, Our GaussTrace. Bottom row, left to right: ResNet (He et al., 2016)+Integrity (Zhang et al., 2020), Points CD (Butt & Maragos, 1998)+Integrity (Zhang et al., 2020). The green edges indicate correct connections with correct directions, pink edges denote correct connections with wrong directions, and red edges represent wrong connections. Missing node represents that this node is absent from the provenance graph constructed by this method. Our method can construct the directional evolutionary relationships among 3DGS models effectively, whereas baselines often fail to infer correct edge directions and frequently introduce spurious connections. For visualization clarity, we display a rendered view of each model, while our analysis is performed on the 3DGS models.

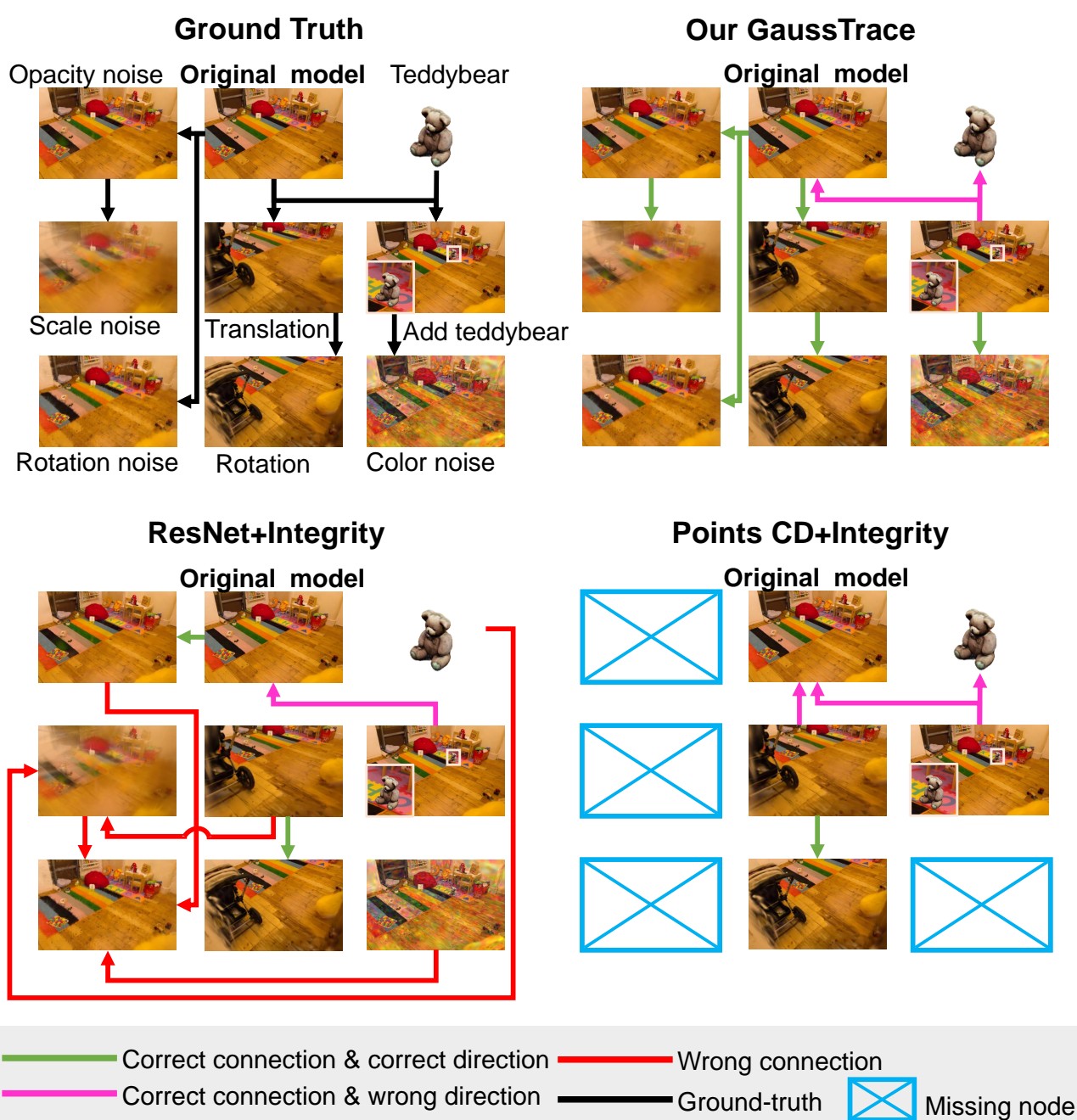

*Figure 8.* Qualitative comparison of 3DGS provenance graph construction. Top row, left to right: ground-truth, Our GaussTrace. Bottom row, left to right: ResNet (He et al., 2016)+Integrity (Zhang et al., 2020), Points CD (Butt & Maragos, 1998)+Integrity (Zhang et al., 2020). The green edges indicate correct connections with correct directions, pink edges denote correct connections with wrong directions, and red edges represent wrong connections. Missing node represents that this node is absent from the provenance graph constructed by this method. Our method can construct the directional evolutionary relationships among 3DGS models effectively, whereas baselines often fail to infer correct edge directions and frequently introduce spurious connections. For visualization clarity, we display a rendered view of each model, while our analysis is performed on the 3DGS models.

