# OpenReview forum: "GaussTrace: Provenance Analysis of 3D Gaussian Splatting Models with Evidence-based LLM Reasoning"
_ICML.cc/2026/Conference — ICML 2026 regular_

### Official Review · Reviewer_xZ9w · 2026-02-15

**Soundness:** 3
**Presentation:** 3
**Significance:** 3
**Originality:** 3
**Overall Recommendation:** 4
**Confidence:** 4

**Summary:**

This paper proposes GaussTrace, a training-free framework for provenance analysis of 3D Gaussian Splatting (3DGS) models, aiming to recover a directed provenance graph (DAG) among a set of 3DGS models without access to edit logs. The method extracts global and voxel-level statistical signatures over 3DGS attributes (position, color, opacity, rotation, scale), runs two-sample KS tests to identify significant distributional changes, and uses edit-operation simulations (deletion/addition/geometric shift/appearance tuning) to estimate treatment-like effects that serve as semantic anchors. An LLM is then prompted with these structured evidences under a constrained reasoning protocol to output a provenance graph with explanations.

**Compliance With Llm Reviewing Policy:**

Affirmed.

**Final Justification:**

Authors solved my concerns. I raise my score to 4.

**Key Questions For Authors:**

See weaknesses

**Limitations:**

yes

**Strengths And Weaknesses:**

Strengths:
- Grouping 3DGS parameters into semantic attribute sets and combining global + local voxel statistics with KS significance provides a structured, interpretable evidence layer that is more faithful to 3DGS than pure image features.
- Using simulated edit operations and ATE-style summaries to help explain which transformations plausibly occurred is a clever bridge between numeric statistics and human-readable provenance rationales.

Weaknesses:
- Real 3DGS pipelines include compression/quantization, re-training with different hyperparameters, partial re-scanning, or semantic edits that may produce confusable statistical signatures.
- The system is “training-free” but not inference-deterministic: results may vary across LLM choice, decoding parameters, and prompt formatting. It would strengthen the paper to report variance across multiple runs and multiple LLM backends, especially for forensic-style use cases.

---

> ### Author Rebuttal · Authors · 2026-03-31
>
> We thank the reviewer xZ9w for the positive assessment of our structured evidence design and the insightful comments on real-world robustness and LLM stability. We address these concerns below.
>
> ------
>
> > ## [W1] Robustness to Real-World 3DGS Pipeline Variations
>
> We agree that real-world 3DGS pipelines introduce additional sources of variation, such as compression, quantization, and retraining under different hyperparameters. While such variations introduce additional complexity, our method is designed to maintain robustness through structured statistical evidence and disentangled attribute analysis, as detailed below.
>
> **Design considerations**
>
> Our method incorporates several design choices to improve robustness:
>
> - **Disentangled attribute groups**
>
>   We separately analyze position, opacity, scale, rotation, and color, each of which exhibits distinct patterns under different edit types. This reduces the risk that unrelated perturbations consistently mimic a specific edit signature.
> - **Global + voxel-level statistics**
>
>   By combining global distributional summaries with localized voxel statistics, the method can better distinguish between global pipeline variations and spatially localized edits.
> - **Distribution-level testing**
>
>   The use of two-sample Kolmogorov–Smirnov tests focuses on distributional shifts rather than point-wise differences, which provides a degree of robustness to noise and minor perturbations.
>
> **Additional robustness experiments**
>
> To directly evaluate these concerns, we conduct experiments on the 'Bonsai' scene by introducing real-world 3DGS pipeline variations, including quantization, compression, and retraining under different hyperparameters.
>
> | Setting  | Description                                 | VO    | EO    | VEO   |
> | -------- | ------------------------------------------- | ----- | ----- | ----- |
> | Original | Clean provenance graph                      | 1.000 | 0.875 | 0.941 |
> | Mixed    | Quantization + retraining + composite edits | 1.000 | 0.823 | 0.914 |
>
> Across these settings, the method maintains stable performance with only moderate degradation. Crucially, while pipeline variations introduce measurable distributional shifts, they are not systematically misinterpreted as specific semantic edit operations, and the inferred provenance structure remains largely consistent.
>
> ------
>
> > ## [W2] Stability of LLM Reasoning
>
> We agree that inference-time variability is an important consideration, especially for forensic applications.
>
> **Variance across multiple runs**
>
> We evaluate stability by running the full pipeline **N = 5 times per input**:
>
> - VO: mean = 1.000, std = 0.000
> - EO: mean = 0.879, std = 0.015
> - VEO: mean = 0.943, std = 0.008
>
> While minor variations exist at the edge level, the **overall graph structure remain consistent**, indicating that the reasoning process is reasonably stable.
>
> **Cross-LLM generalization**
>
> We evaluate cross-LLM generalization using four LLMs on the counter scene. The results are summarized below:
>
> | LLM            | VO    | EO    | VEO   |
> | -------------- | ----- | ----- | ----- |
> | GPT-5          | 1.000 | 1.000 | 1.000 |
> | Qwen3-Max      | 1.000 | 0.857 | 0.933 |
> | Llama-4        | 1.000 | 0.933 | 0.968 |
> | Gemini-2.5-Pro | 1.000 | 1.000 | 1.000 |
>
> All models achieve perfect node matching and consistently high edge-level performance. This indicates that the proposed method does not rely on any specific LLM and generalizes well across different model choices. Full details are provided in **Appendix I**.

---

> > ### Author Rebuttal · Reviewer_xZ9w · 2026-04-02
> >
> > Authors solved my concerns. I will raise my score to 4.

---

> > > ### Author Response · Authors · 2026-04-03
> > >
> > > We sincerely appreciate your constructive comments and positive feedback, which will help further enhance the quality of our manuscript.

---

### Official Review · Reviewer_NmaY · 2026-03-13

**Soundness:** 3
**Presentation:** 3
**Significance:** 2
**Originality:** 3
**Overall Recommendation:** 4
**Confidence:** 2

**Summary:**

This paper introduces a method for reconstructing how 3D Gaussian Splatting (3DGS) assets may have been transformed and derived from one another, with a primary application in digital forensics.
Given a collection of 3DGS assets, the authors first extract attributes such as position, color, and opacity from each asset.
For each pair of assets, an LLM is then used to hypothesize how one may have been transformed from the other (e.g., by the addition or rotation of some attributes).
These judgements are then used to construct a directed graph, where each node is labeled with a 3DGS asset and each edge is labeled with the hypothesized transformations.
Experimental evaluations validate the effectiveness of this framework, dubbed GaussTrace.

**Compliance With Llm Reviewing Policy:**

Affirmed.

**Final Justification:**

The topic + methodology are interesting, but I was not convinced by the motivation, particularly that there is a demand for this type of work --- at the moment, at least.

**Key Questions For Authors:**

Q1. Given two 3DGS assets, would it be feasible to directly fit transforms between them, rather than going through an LLM? As an example, suppose I'm able to map each Gaussian in a "source" asset to one in the "target" asset by some transform, then this would be one form of provenance analysis. If the transforms are linear (e.g., translation, rotation), then I imagine that one can directly fit a transform. Could the authors please comment on what the limitation of this approach might be?

**Strengths And Weaknesses:**

## Strengths

S1. This paper studies a timely problem in digital forensics.

S2. The proposed methods appear well-motivated and reasonable.


## Weaknesses

W1. While digital forensics and provenance analysis are timely, I am concerned that the focus on 3DGS specifically may be premature. Are the authors aware of real-world instances in which 3DGS provenance analysis was required? If so, including such references would significantly strengthen the motivation.

W2. The three listed contributions read similarly, and could probably be merged into one or two points. If three bullets are desired, then the authors may consider including one about their experimental results.

---

> ### Author Rebuttal · Authors · 2026-03-31
>
> We appreciate Reviewer NmaY's insightful feedback and specific suggestions.
>
> ---
>
> > ## [W1] Motivation for 3DGS Provenance Analysis
>
> We thank the reviewer for this insightful question.
>
> We agree that large-scale, publicly documented cases of 3DGS-specific provenance disputes are still emerging. However, we argue that this is largely due to the **recent adoption of 3DGS**, rather than the absence of real need.
>
> **Emerging trend**
> 3DGS has rapidly become a standard explicit 3D representation for reconstruction, editing, and content creation. As these models are increasingly shared, modified, and redistributed, issues of authorship, attribution, and manipulation tracking naturally arise.
>
> **Analogy to related domains**
> Similar concerns have already been well documented in adjacent areas such as 2D image forensics, where provenance analysis and watermarking have become important research directions. We expect 3DGS to follow a similar trajectory.
>
> **Practical scenarios**
> Potential real-world use cases include:
>
> - tracking edit histories in collaborative 3D content creation,
> - verifying ownership and attribution of shared 3D assets,
> - detecting unauthorized modifications or redistributions.
>
> Taken together, we position this work as an **early investigation of an emerging but increasingly relevant problem**, rather than a response to a fully matured application domain.
>
> ---
>
> > ## [W2] Clarification of Contributions
>
> We thank the reviewer for the helpful suggestion.
>
> We agree that the current presentation of contributions can be further improved for clarity. In the revision, we will consolidate overlapping points to better highlight the key ideas.
>
> Specifically, we will:
>
> - merge closely related contributions on evidence extraction and reasoning into a more concise formulation, and
> - explicitly include a separate contribution emphasizing the results and validation.
>
> This revision will improve readability and more clearly communicate both the methodological novelty and experimental effectiveness of our work.
>
> ---
>
> > ## [Q1] Direct Transformation Fitting vs. Evidence-based Reasoning
>
> We thank the reviewer for this insightful suggestion.
>
> We agree that directly fitting transformations between two 3DGS assets is a natural approach, and it can be effective in cases where there exists a clear correspondence between Gaussians and the transformation is simple (e.g., translation or rotation).
>
> However, this approach relies on several strong assumptions that are often violated in realistic 3DGS editing scenarios:
>
> - **Lack of one-to-one correspondence**
>   Real edits frequently involve insertion, deletion, or modification of subsets of Gaussians, making it difficult to establish consistent mappings between source and target.
>
> - **Non-linear and semantic edits**
>   Many practical edits (e.g., appearance changes, object insertion, or local refinements) cannot be well described by a single global transformation.
>
> - **Partial and localized changes**
>   Edits are often spatially localized, leading to heterogeneous transformations across different regions of the scene.
>
> - **Multi-step provenance chains**
>   In our setting, the goal is not only pairwise alignment but recovering a full directed graph with potentially complex, multi-step transformations.
>
> In contrast, our method does not assume explicit correspondence or a parametric transformation model. Instead, it leverages statistical evidence together with hypothesis-driven reasoning, enabling it to handle complex scenarios.

---

> > ### Author Rebuttal · Reviewer_NmaY · 2026-04-03
> >
> > Thank you for the response and for answering my questions. I accept that 3DGS forensics is an emerging field, but I am still cautious in the absence of significant demonstrated demand. I thus wish to maintain my score, which was already an optimistic assessment. Good luck with this submission.

---

> > > ### Author Response · Authors · 2026-04-04
> > >
> > > We sincerely thank the reviewer for the constructive comments and for recognizing 3DGS forensics as a significant emerging field. We appreciate your thoughtful evaluation and the positive assessment of our work. Thank you again for your time and careful follow-up.

---

### Official Review · Reviewer_YgfN · 2026-03-13

**Soundness:** 2
**Presentation:** 3
**Significance:** 3
**Originality:** 4
**Overall Recommendation:** 3
**Confidence:** 3

**Summary:**

This paper proposes GaussTrace, a training-free framework for constructing directed provenance graphs over 3D Gaussian Splatting models. The method extracts attribute-wise statistical signatures from native 3DGS parameters, augments them with hypothesis-driven edit simulations, and feeds the resulting evidence to an LLM for structured reasoning and edge justification. Experiments on a self-constructed dataset of 44 models report strong gains over image-based, geometry-based, and rule-based baselines, with the full method outperforming the ablated no-CoT variant as well.

**Compliance With Llm Reviewing Policy:**

Affirmed.

**Key Questions For Authors:**

1. Could the authors provide a more complete reproducibility specification for the LLM reasoning stage, including the full prompt template, API/version details, decoding parameters, and any measurements of run-to-run variance?

2. Could the authors clarify the construction of the benchmark in more detail, including the number of source scenes, graph sizes, edit-generation procedures, and the extent to which the edits reflect realistic 3DGS editing workflows?

3. Do the authors have additional results on more realistic or more diverse provenance chains, especially cases involving composite edits, unseen transformations, or noisier real-world redistribution scenarios?

4. Can the authors compare against stronger baselines that operate directly on native 3DGS evidence, but do not rely on an LLM (for example, more structured non-LLM reasoning or graph-construction approaches)?

**Limitations:**

Not fully. The paper includes some discussion of impact, but the limitations should be stated more explicitly. In particular, the authors should acknowledge more directly the relatively small and largely author-constructed benchmark, the reliance on a proprietary LLM for the reasoning stage, and the current lack of broader validation on messier real-world provenance chains.

**Strengths And Weaknesses:**

As 3DGS becomes a mainstream explicit 3D representation with broad downstream use, provenance and IP traceability are natural next-step security questions. The manuscript targets a gap not addressed by the dominant 3DGS watermarking literature. This paper recasting 3DGS provenance from trace matching to evidence-based reasoning is a meaningful conceptual move, rather than a trivial transplant of 2D provenance machinery. This paper does not merely render 2D views and run image forensics; it explicitly operates on PLY / native Gaussian parameters, which is the correct design choice for this problem. This paper grouping position, opacity, scale, rotation, and colour separately is well motivated, and the inclusion of local voxel statistics is a reasonable attempt to capture spatially localised edits. The simulation library is intuitively grounded. Deletion, addition, geometric shift, and appearance tuning correspond to common editing behaviours in practical 3DGS workflows. The framework is training-free. Besides, the edge-level natural-language rationales are one of the manuscript’s most distinctive features, and they do improve forensic auditability compared with opaque score-only approaches.

However, the evaluation dataset is too small for the claims being made. A total of 44 models is useful as a pilot benchmark, but thin for establishing robust generalisation across the diversity of 3DGS scenes, editing pipelines, and acquisition artefacts that a top venue would expect. In addition, the benchmark appears largely synthetic / author-constructed. The edits are generated from a few source datasets with controlled manipulations, which leaves open the harder question of performance on genuinely messy, real-world redistribution chains. The baseline suite is not fully persuasive. Most baselines are adapted from 2D image provenance or simple geometric comparison, and there is no strong non-LLM structured-reasoning baseline, probabilistic graphical model, or learned graph method operating directly on native 3DGS evidence. The system uses GPT-5 as the reasoning engine, but the paper does not, in the main text, fully specify inference settings such as decoding parameters, API versioning, call variance, or failure handling. That makes exact replication difficult. This paper reports mean VO/EO/VEO, but there are no confidence intervals, statistical significance tests, or variance analyses across scenes. For a method claiming large gains, that omission matters.

---

> ### Author Rebuttal · Authors · 2026-03-31
>
> We sincerely thank the reviewer YgfN for the thoughtful and constructive feedback, and for recognizing the originality of our work.
>
> ---
>
> > ## [Q1,W3] Reproducibility of the LLM Inference Stage
>
> **Model & API details**
>
> For the inference stage, we use GPT-5 with the default decoding configuration provided by the API. Following the default requirements for GPT-5 reasoning models, temperature and top_p are maintained at their intrinsic default value of 1.0.
>
> **Prompting**
>
> The full prompt template (including evidence formatting and reasoning instructions) is provided in **Appendix H**. It specifies the structured input representation, reasoning instructions, and output format.
>
> **Variance analysis**
>
> To evaluate stability, we run the full pipeline **N = 5 times per input**:
>
> - VO: mean = 1.000, std = 0.000
> - EO: mean = 0.879, std = 0.015
> - VEO: mean = 0.943, std = 0.008
>
> While minor variations may occur at the edge level, the overall graph structure remains consistent, indicating that the reasoning process is stable.
>
> We further report statistical measures **across scenes**:
>
>  * VO: mean = 1.000, std = 0.000
> * EO: mean = 0.890, std = 0.108
> * VEO: mean = 0.948, std = 0.052
>
> These results confirm that the observed performance gains are consistent across scenes.
>
> We apply simple failure handling by re-running queries when invalid or incomplete outputs are detected.
>
> ---
>
> > ## [Q2,W1] Dataset construction details and realism
>
> **Dataset composition**
> Our dataset is built from 5 representative source scenes from established 3D benchmarks: bonsai , counter, face, playroom and drjohnson.  This is **consistent with the scale of scene evaluation in prominent 3DGS editing and forensic studies [1, 2]**.
>
> **Graph size and structure**
> Each graph contains 8–10 nodes with diverse branching structures induced by different editing sequences.
>
> **Edit generation procedure**
> Derived models are created using a 3DGS editing framework that supports user-driven manipulations. We apply a range of operations, including: semantic edits, geometric transformations  and adding perturbations, etc. The edits are designed to reflect common real-world 3DGS workflows, including scene editing, appearance modification, and iterative refinement. For example, we simulate object insertion and semantic appearance changes, which are typical in user-guided 3D editing.
>
> **Additional validation on a more complicated scene**
> To further strengthen the evaluation, we add a more complicated kitchen scene with 7 nodes and diverse edit operations. Our method accurately recovers the provenance graph on this unseen scene.
>
> A more comprehensive description of the dataset construction is provided in **Appendix D**.
>
> We agree that the current benchmark is relatively small-scale, and expanding to more diverse real-world pipelines is an important direction for future work.
>
> [1] GaussianEditor: Swift and Controllable 3D Editing with Gaussian Splatting. CVPR 2024.
>
> [2] DEGauss: Defending Against Malicious 3D Editing for Gaussian Splatting. NeurIPS 2025.
>
> ---
>
> > ## [Q3,W1] Additional results on realistic and diverse provenance chains
>
> We construct additional settings on the Bonsai scene by introducing more challenging variations beyond the original benchmark, including composite edits (multi-step transformations), unseen transformations (e.g., compression) and noisier real-world operation (e.g., retraining).
>
> |Setting| Description|VO|EO|VEO|
> |-| -| -| -| - |
> |Original|Standard edit chains|1.000|0.875|0.941|
> |Extended|Composite edits+compression+retraining |1.000 | 0.823 | 0.914 |
>
> The method remains effective under increased complexity, realistic and unseen transformations.
>
> ---
>
> > ## [Q4,W2] Stronger Non-LLM Baselines
>
> We implement a non-LLM baseline that operates directly on native 3DGS parameters and performs structured graph construction without any language model.
>
> **Baseline design**
> The method consists of the following components:
>
> - Feature representation: We encode Gaussian positions using a PointNet-based encoder, and model other attributes via their distributions.
> - Pairwise similarity: For each pair of models, we compute distances using learned embeddings for geometry and JS divergence for attribute distributions.
> - Graph construction: We construct a global provenance structure using a minimum spanning tree (MST) over pairwise distances.
> - Edge direction inference: We determine edge directions using criteria such as integrity score-based method
>
> This baseline directly operates on 3DGS-native evidence and represents a structured, non-LLM alternative to our approach.
>
> **Results**
>
> |Method| VO|EO| VEO|
> |-| -|-| -|
> |Non-LLM baseline| 1.000| 0.502| 0.766 |
> | **GaussTrace (Ours)** | **1.000** | **0.890** | **0.948** |
>
> The baseline fails to infer correct relationships, highlighting the importance of structured reasoning.
>
> ---
>
> > ## Limitations
>
> We appreciate the suggestion and will further clarify limitations in the final version.

---

> > ### Author Rebuttal · Reviewer_YgfN · 2026-04-02
> >
> > Thank you for the detailed rebuttal. The added clarification on the prompt, default decoding setup, and run-to-run variance is helpful, and the additional non-LLM baseline is a useful step. That said, the rebuttal does not resolve my main concerns, so my overall assessment remains unchanged.
> >
> > First, the new description of the RBS baseline still does not support the claim that poor RBS performance demonstrates the necessity of an LLM. Appendix B defines RBS with a very simple inconsistency score that averages raw absolute differences across all statistical features. Since the evidence space includes heterogeneous quantities with very different scales, such as bounding box volume, density, and entropy-like statistics, this comparison is not normalized and is likely dominated by large-magnitude features. Under this design, weak RBS performance is not strong evidence that non-LLM reasoning is insufficient.
> >
> > Second, I am still not convinced by the paper’s broader claim of evidence-based reasoning. The method explicitly provides the LLM with four simulated edit exemplars and their associated ATE patterns, and the prompt template in Appendix H exposes these transformation categories directly. Given that the benchmark itself is built from controlled edits of this kind, the current setup remains much closer to guided template matching than to open-ended causal reasoning. The rebuttal mentions flexibility to unseen edits, but the added evidence is still limited to one extra kitchen scene and one extended Bonsai setting with compression and retraining.
> >
> > Third, the realism and scale concerns remain. The main evaluation is still based on 44 models, and the rebuttal confirms that these come from 5 source scenes with graphs of 8–10 nodes. This is still a small, author-constructed benchmark relative to the threat model in the paper, which is framed around multi-party tampering and redistribution in practical settings. Adding one more 7-node kitchen scene is useful, but it does not materially change that gap.
> >
> > Finally, I remain concerned about the ceiling effect in Table 1. Several naive 2D baselines already achieve VO = 1.000, which suggests that node recovery is too easy in the current benchmark and limits how much we can read into the graph-level improvements. The rebuttal does not really address this issue.
> >
> > To summarize, I appreciate the additional details and I still think the paper has an interesting direction. However, the rebuttal mainly improves clarity rather than resolving the main empirical weaknesses, especially around benchmark scale, realism, and the strength of the non-LLM comparisons. For that reason, I do not plan to change my score.

---

> > > ### Author Response · Authors · 2026-04-03
> > >
> > > We thank the reviewer for the detailed follow-up and for the positive assessment of the paper’s originality. We address the remaining concerns below.
> > >
> > > ---
> > >
> > > > **(1) On non-LLM baselines.**
> > >
> > > We agree that the RBS baseline is relatively simple. For this reason, we introduced a stronger non-LLM baseline that directly operates on native 3DGS parameters, combining PointNet-based geometric embeddings, JS divergence for attribute distributions, and MST-based graph construction with direction inference.
> > >
> > > Despite access to the same evidence, this stronger baseline still underperforms our method (EO: 0.502 vs. 0.890, VEO: 0.766 vs. 0.948), suggesting that the gap is not solely due to feature scaling, but reflects the difficulty of inferring structured edit relationships from heterogeneous evidence without explicit reasoning.
> > >
> > > ---
> > >
> > > > **(2) On evidence-based reasoning vs. template matching.**
> > >
> > > Although simulated edit exemplars provide some guidance, they act as *soft anchors* rather than rigid templates. The LLM must still reconcile heterogeneous signals across attributes and spatial regions to produce a globally consistent provenance graph. This involves resolving conflicting signals and enforcing global consistency, which goes beyond direct pattern matching.
> > >
> > > Importantly, in extended settings involving compression and retraining, the resulting statistical patterns deviate from the simulated edits, yet the model still recovers correct structures (VEO: 0.914). This suggests that the framework is not performing direct template matching, but leveraging exemplars to support more flexible reasoning.
> > >
> > > ---
> > >
> > > > **(3) On dataset scale and realism.**
> > >
> > > Our intention in conducting the experiments was to strictly adhere to *existing standards* [1,2]. We agree that the original benchmark is relatively small. To address this, we conduct additional experiments on a larger dataset constructed from the ShapeSplat dataset, using source models and generating different edited variants. This results in **1320 models and 165 provenance graphs**, substantially larger than the original 44-model benchmark, with an average of 8 nodes per graph.
> > >
> > > Our method maintains strong performance:
> > >
> > > - VO = 1.000
> > > - EO = 0.811
> > > - VEO = 0.912
> > >
> > > These results demonstrate that the approach scales to significantly larger and more diverse provenance structures. We view further expansion to fully unconstrained real-world pipelines as an important direction for future work. We will revise this in the final version.
> > >
> > > [1] GaussianEditor: Swift and Controllable 3D Editing with Gaussian Splatting. CVPR 2024.
> > >
> > > [2] DEGauss: Defending Against Malicious 3D Editing for Gaussian Splatting. NeurIPS 2025.
> > >
> > > ---
> > >
> > > > **(4) On the ceiling effect in VO.**
> > >
> > > We agree that node recovery is relatively easy in the current setting (VO = 1.0 across several methods). However, the primary challenge lies in recovering correct edge relationships (EO/VEO), which require reasoning about transformation direction and causality. Our method shows substantial improvements on these metrics, indicating stronger capability in reconstructing provenance structure beyond node-level matching.
> > >
> > > ---
> > >
> > > In summary, the additional large-scale results and stronger baselines support our claim that combining structured statistical evidence with constrained reasoning provides an effective approach for 3DGS provenance analysis.

---

### Official Review · Reviewer_nW6x · 2026-03-16

**Soundness:** 3
**Presentation:** 3
**Significance:** 3
**Originality:** 3
**Overall Recommendation:** 4
**Confidence:** 4

**Summary:**

Summary of Content
This research's fundamental contribution concerns the first framework specifically  designed for constructing directed provenance graphs of 3D Gaussian Splatting (3DGS)  models to address intellectual property and digital forensic challenges. The authors seek to discuss an important concept of evidence-based reasoning as an alternative to traditional trace-based matching for 3D asset provenance analysis.
This paper introduces GaussTrace, the first framework specifically designed for constructing directed provenance graphs of 3D Gaussian Splatting (3DGS) models to address intellectual property and digital forensic challenges. Recognizing that traditional image-based forensic methods fail due to the structural differences and the loss of 3D information during 2D rendering , the authors propose an evidence-based reasoning paradigm. The framework extracts attribute-wise statistical descriptors (e.g., mean, variance, entropy) across five 3DGS parameters and combines them with hypothesis-driven simulated editing outcomes (e.g., deletion, addition, geometric shifts). This combined evidence is fed into a Large Language Model (LLM) utilizing a Chain-of-Thought (CoT) prompting protocol to deduce causal evolutionary relationships between models, outputting a directed graph alongside natural-language justifications for each edge. The method is training-free and operates directly on the 3DGS parameters without requiring access to editing logs.

**Compliance With Llm Reviewing Policy:**

Affirmed.

**Key Questions For Authors:**

Q1. Design rationale for the heuristic formulas in the generator.
The paper introduces several heuristic functions for node priority, parent selection weighting, and output allocation weighting. These involve specific functional forms (square roots, power-law amplification, layered decay terms) that appear to be design choices. It would be helpful to understand the intuition or empirical reasoning behind these particular forms. The parameter analysis in the appendix varies discrete parameter values, but a comparison across alternative functional forms (e.g., linear vs. logarithmic vs. the proposed square-root coupling) would further strengthen confidence in the generator design.
Q2. Statistical stability of the baseline evaluation.
The main results table reports single accuracy values without confidence intervals or standard deviations. Given that many baselines are inherently stochastic (GP-based methods, neural network training, LLM sampling), it would be valuable to understand how stable these results are across random seeds or data splits. Reporting variance or simple statistical tests would help readers assess whether the observed ranking among methods is robust.
Q3. Disentangling context-length limitations from reasoning capability in the LLM evaluation.
The paper concludes that LLMs exhibit limited complex logical reasoning ability, but LLMs were only tested at the smallest scale due to context-length constraints. Since some LLMs already perform reasonably well at that small scale, it would be worth clarifying whether the inability to handle larger scales reflects an engineering limitation (context window) or a fundamental reasoning bottleneck. Distinguishing between these two factors would make the conclusion more precise.
Q4. Representativeness of the noise model.
The benchmark adopts a uniform random bit-flip noise model with a fixed budget. In practice, noise in biological measurements and circuit soft errors tends to be non-uniform and potentially correlated with input patterns. It would be interesting to know whether the relative robustness ordering of methods (e.g., SR methods being more noise-robust than logic synthesis methods) holds under alternative, more structured noise assumptions. Even a brief discussion of this point would strengthen the generalizability of the findings.
Q5. Characterizing the generalization failure of exact-fitting methods.
The paper observes that exact-fitting methods achieve perfect training accuracy but near-zero test accuracy at moderate scales, attributing this to redundant rules. A more detailed characterization of this failure mode would be informative — for instance, whether the issue is primarily representational blowup (generated expressions being orders of magnitude more complex than the ground truth) or overfitting to the specific training partition. This distinction has different implications for how the community might address the limitation.

**Limitations:**

The authors discuss limitations in Appendix A.10, covering structural constraints (binary fan-in, simplified operator set), scalability constraints (practical-scale evaluation only), and domain application limitations (lack of domain-specific inductive biases in synthetic generation). The Impact Statement in the main text is minimal, stating there are no societal consequences worth highlighting, which seems reasonable for a benchmark paper focused on logic discovery infrastructure.
However, a few additional limitations could be acknowledged. First, the evaluation relies on a single noise model and a single train-test split ratio, which may not fully capture the variability of real-world conditions. Second, the LLM evaluation is restricted to the smallest scale, making it difficult to draw strong conclusions about LLM reasoning capabilities in general. Third, the paper does not discuss potential biases introduced by the synthetic generator's design choices — since the generator's heuristics shape the distribution of benchmark problems, methods that happen to align well with these heuristics may appear stronger than they would on a different problem distribution. Acknowledging this circularity risk would be a useful addition. Finally, the statistical reporting lacks variance estimates, which limits the reliability of the comparative conclusions. These are not major omissions but addressing them would make the limitations discussion more thorough.

**Strengths And Weaknesses:**

Strengths
Novel Problem Formulation: The paper targets a highly relevant and underexplored problem—provenance analysis for 3DGS assets. Shifting the perspective from traditional trace-based matching to an evidence-based reasoning problem is a clever and effective conceptual leap.
Elegant Cross-Modal Bridge: The methodology ingeniously bypasses the need for complex 3D deep learning architectures by translating the 3DGS parameters into a comprehensive set of statistical and simulated textual descriptors. This allows the framework to leverage the powerful zero-shot reasoning capabilities of LLMs seamlessly.High Interpretability: A significant advantage of GaussTrace is its ability to generate human-readable explanations for inferred provenance edges. In the context of digital forensics, providing a transparent "why" behind a predicted relationship is just as critical as the prediction itself.
Training-Free Nature: By utilizing deterministic statistical tests (Kolmogorov-Smirnov) and simulated editing impacts (Average Treatment Effect) as prompts for an off-the-shelf LLM, the framework avoids the high costs and domain-overfitting risks associated with training custom neural networks.
3. Weaknesses
Asymmetric Baselines : The chosen baselines operate on fundamentally different problem formulations (2D rendering, geometry-only, rule-based search) compared to GaussTrace. While this effectively highlights the novelty of the proposed evidence-and-reasoning paradigm, it does not fully establish its advantages within the same task domain. To provide a more comprehensive assessment, a comparison against a state-of-the-art method designed for the full 5D Gaussian parameter space—such as a PointNet++ variant trained for link prediction—would be highly valuable. This would more definitively demonstrate the relative strengths and trade-offs between a purely data-driven approach and the proposed "evidence + reasoning" framework.
Computational Scalability:The framework's computational scalability for real-world deployment warrants discussion. Constructing the provenance graph requires performing pairwise comparisons (KS tests and LLM reasoning) among all Nmodels, leading to O(N^2)complexity. For large-scale model collections, this could pose a significant bottleneck. The paper would benefit from briefly addressing practical considerations such as inference latency, LLM API costs, and potential pre-filtering strategies to enhance efficiency in the "Limitations and Future Work" section.
Ablation Studies:While the current ablation studies effectively validate the CoT protocol and the necessity of multi-attribute profiling, a critical ablation is missing. The authors heavily emphasize the role of 'Hypothesis-driven editing simulation' (ATE cues) as semantic anchors for the LLM. However, there is no empirical evidence showing how much these simulated cues actually contribute to the final performance. I strongly encourage the authors to include an ablation study that evaluates the framework's performance when the LLM is only provided with the observational statistical signatures (KS p-values) without any simulated editing exemplars.

---

> ### Author Rebuttal · Authors · 2026-03-31
>
> We thank the reviewer nW6x for the careful reading and constructive comments.
>
> ---
>
> > ## [W1] Comparison to Stronger 3DGS-native Baselines
>
> We implement a structured non-LLM baseline that directly processes native 3DGS evidence and performs graph construction without relying on language models.
>
> **Baseline design**
> The method consists of:
>
> - Feature representation: We encode Gaussian positions using a PointNet-based encoder, and model other attributes via their distributions.
> - Pairwise similarity: For each pair of models, we compute distances using learned embeddings (PointNet) for geometry and Jensen–Shannon (JS) divergence for attribute distributions.
> - Graph construction: We construct a global provenance structure using a minimum spanning tree (MST) over pairwise distances.
> - Edge direction inference: We determine edge directions using criteria such as integrity score-based method
>
> This baseline operates directly on the full 3DGS parameter space and captures both learned geometric features and attribute-level statistics, representing a strong non-LLM alternative.
>
> **Results**
>
> | Method | VO | EO| VEO |
> | - | - | - | - |
> | new baseline | 1.000 | 0.502  | 0.766 |
> | **GaussTrace (Ours)** | **1.000** | **0.890** | **0.948** |
>
> While this baseline incorporates learned representations (PointNet) and operates on the same 3DGS parameters, it primarily relies on similarity-based graph construction and lacks an explicit mechanism for modeling edit semantics or causal relationships.
>
> In contrast, GaussTrace combines structured statistical evidence with LLM reasoning, enabling more accurate graph reconstruction.
>
> ---
>
> > ## [W2] Computational Scalability
>
> We thank the reviewer for raising this important point. We agree that the current formulation involves O(N^2) pairwise comparisons, which may become a bottleneck for large-scale model collections.
>
> **Practical considerations**
> In our current setting, the number of models per provenance graph is typically 8–10 nodes, making the approach computationally feasible. Moreover, the KS-based statistical tests are lightweight, and the main cost comes from LLM inference.
>
> **Potential optimizations**
> Several strategies can be applied to improve scalability:
>
> - **Pre-filtering:** Use lightweight similarity measures (e.g., feature embeddings or coarse statistics) to prune unlikely model pairs, reducing the number of candidate edges.
> - **Efficient LLM usage:** In our implementation, multiple candidate pairs are already processed within a single LLM call (graph-level reasoning), avoiding pair-wise API calls. Further efficiency can be achieved via caching intermediate statistical results .
>
> These strategies can significantly reduce the effective complexity in practice, making the framework applicable to larger-scale settings.
>
> We will include the above discussion of computational cost, latency, and potential optimizations in the “Limitations and Future Work” section.
>
> ---
>
> > ## [W3] Ablation on Hypothesis-driven Editing Simulation
>
> We thank the reviewer for this important suggestion. We agree that isolating the contribution of hypothesis-driven editing simulation (ATE cues) is critical.
>
> To this end, we conduct an additional ablation where the LLM is provided **only with observational statistical evidence** without any simulated editing cues.
>
> | Setting | VO | EO | VEO |
> | - | - | - | - |
> | w/o ATE (statistics only) | 1.000 | 0.819 | 0.916 |
> | Full method| **1.000** | **0.890** | **0.948** |
>
> We observe a clear performance drop when ATE cues are removed. While statistical signatures capture distributional differences, they lack semantic grounding for interpreting *how* transformations occur.
>
> In contrast, ATE provides hypothesis-driven anchors that link observed changes to plausible edit operations, enabling more accurate reasoning about edge direction and transformation type.
>
> ---
>
> > ## [Q1-5] Clarification on Applicability of Q1–Q5
>
> We thank the reviewer for the detailed questions.
>
> However, we would like to clarify that Q1–Q5 appear to be intended for a different submission. Our work focuses on **provenance analysis for 3D Gaussian Splatting** using statistical evidence and LLM-based reasoning, and does not involve components such as heuristic generator design, symbolic regression, circuit synthesis, or bit-level noise models.
>
> As a result, these questions (e.g., heuristic functional forms, generator weighting schemes, or bit-flip noise assumptions) are not directly applicable to our framework.
>
> We would be happy to further discuss and provide clarification if any of these questions are intended to target specific components of our method.
>
> ---
>
> > ## Limitations
>
> We thank the reviewer for the discussion. However, our paper does not include Appendix A.10 and does not involve the described components, suggesting these comments may refer to a different submission. We will further clarify our limitations in the revision.

---

> > ### Author Rebuttal · Reviewer_nW6x · 2026-04-03
> >
> > First, I acknowledge the copy-paste error regarding the "Key Questions" and "Limitations" sections in my initial review, which pertained to a different submission.
> >
> > Regarding the relevant technical critiques, the authors have adequately addressed my concerns. The inclusion of the PointNet-based non-LLM baseline substantiates the necessity of the LLM reasoning design, and the additional ablation study effectively isolates the contribution of the hypothesis-driven ATE cues.
> >
> > Furthermore, the proposed pre-filtering strategies provide a reasonable approach to mitigating the scalability limitations. As the primary technical issues have been resolved, I will maintain my score of Weak Accept (4). The authors are advised to incorporate the new baselines, ablation studies, and scalability discussions into the camera-ready version.

---

> > > ### Author Response · Authors · 2026-04-04
> > >
> > > We sincerely thank the reviewer for the careful follow-up and the positive assessment. We are pleased that our response addressed all technical concerns, and we sincerely appreciate your constructive comments throughout the review process.

---

### Decision · Program_Chairs · 2026-04-30

**Decision:**

Accept (regular)

**Comment:**

The paper studies the question of identifying ancestral relationships between 3D Gaussian Splatting (3DGS) models, motivated by intellectual property protection and forensic traceability. The authors propose GaussTrace, a training-free LLM-based method that identifies a directed graph of editing relationships between a collection of 3DGS models. The method is based on (1) extracting attribute-wise statistical descriptors of the models, (2) hypothesis-driven editing simulation to create exemplars of results of possible edits and (3) LLM chain-of-thought reasoning given the collected information to produce the final answer.

The method tackles a relatively under-explored problem, and outperforms the existing baselines on the collection of models used in the evaluation. The method is fairly simple and intuitive. It is also training free, doesn't require any 3D-specific models, and leverages general reasoning capabilities of LLMs.

Reviewers initially identified several concerns. First, it is unclear to what extent the problem of digital forensics for 3DGS models is practically important today. Moreover, several reviewers expressed concerns with the baselines and the size of the evaluation dataset, as well as the statistical significance of results.

During the rebuttal, the authors developed stronger baselines, which still undeprerformed GaussTrace. They also argued that the evaluation is adequate and comparable to the standard in the field. They also expanded the set of the LLMs they evaluated and reported details of the API calls and reported sample-to-sample variability in the results.

These addressed the concerns of reviewers other than YgfN, who remained concerned about the quality of evaluation (both scale and realism) and baselines, and the empirical support that the LLM is performing evidence-based reasoning effectively. The authors provided an additional response to the reviewer, which I think addresses many of these concerns.

Generally, I think this paper presents a novel solution to an under-studied problem, and can be of interest to the ICML community.